# Avoiding spurious sharpness minimization broadens applicability of SAM

**Sidak Pal Singh** [1]   **Hossein Mobahi** [2]   **Atish Agarwala** [2]   **Yann Dauphin** [2]

## Abstract

Curvature regularization techniques like Sharpness Aware Minimization (SAM) have shown great promise in improving generalization on vision tasks. However, we find that SAM performs poorly in domains like natural language processing (NLP), often degrading performance — even with twice the compute budget. We investigate the discrepancy across domains and find that *in the NLP setting,* SAM *is dominated by regularization of the logit statistics —— instead of improving the geometry of the function itself.* We use this observation to develop an alternative algorithm we call FUNCTIONAL-SAM, which regularizes curvature only through modification of the statistics of the overall function implemented by the neural network, and avoids spurious minimization through logit manipulation. Furthermore, we argue that preconditioning the SAM perturbation also prevents spurious minimization, and when combined with FUNCTIONAL-SAM, it gives further improvements. Our proposed algorithms show improved performance over ADAMW and SAM baselines when trained for an equal number of steps, in both fixed-length and Chinchilla-style training settings, at various model scales (including *billion-parameter scale*). On the whole, our work highlights the importance of more precise characterizations of sharpness in broadening the applicability of curvature regularization to large language models (LLMs).

## 1. Introduction

One of the most fundamental questions in machine learning research is: how do we train models that are useful beyond their training data? This question arises in multiple scenarios — from generalizing to unseen samples, dealing with distribution shift, and fine-tuning on specific domains. A commonly held belief is that it is important for models to converge to *well-behaved* and *robust* solutions. The 'regularity' of the model is often obtained using *regularization* techniques, which — even in the day and age of LLMs — remain an indispensable part of any training algorithm.

Some of the most prominent regularization methods include weight decay (Krogh & Hertz, 1991), dropout (Srivastava et al., 2014), data augmentation (Ciregan et al., 2012; Krizhevsky et al., 2012), Mixup (Zhang, 2017), and curvature-based controls (Foret et al., 2020; Wu et al., 2020). In recent years, *curvature regularization* techniques have gained popularity due to their effectiveness in promoting generalization. These techniques bias learning dynamics to areas of lower curvature (i.e., less sharp regions) in the loss landscape (Chaudhari et al., 2017; Keskar et al., 2017; Foret et al., 2020; Pittorino et al., 2020; Wu et al., 2020). The origins of these curvature or *sharpness minimization* techniques can be traced back to the classical ideas of minimum description length (Rissanen, 1978; Hinton & Van Camp, 1993; Hochreiter & Schmidhuber, 1997). Lately, their development has been inspired by their success in a large-scale correlational study (Jiang et al., 2019) and in the NeurIPS generalization competition (Jiang et al., 2020).

Sharpness minimization has demonstrated significant improvement on vision tasks. In contrast, there has not been much uptake of these methods in NLP[1] and, especially for as cornerstone (Brown et al., 2020) a task in NLP as language modeling. Curiously enough, we observe that Sharpness Aware Minimization (SAM), one of the best studied sharpness regularization methods (Foret et al., 2020), shows poor performance here; indeed, its validation metrics are typically worse than ADAMW throughout training (Figure 1), despite using more computation per step. Hence, this raises the following questions which form the basis of our work:

> *What are the reasons for* SAM*'s poor performance in language modeling? How can they be addressed to successfully yield generalization benefits, while being equal in cost as* SAM*?*

Towards this end, in **Section 3**, we perform *a novel analysis*

---

[1]Google Research [2]Google DeepMind. Correspondence to: Sidak Pal Singh <ssidak@google.com>.

*Proceedings of the 42nd International Conference on Machine Learning*, Vancouver, Canada. PMLR 267, 2025. Copyright 2025 by the author(s).

---

[1]The most notable exception of SAM in NLP is in the fine-tuning scenario (Bahri et al., 2022), where the parameters are constrained to move smaller distances by the very nature of the setup.

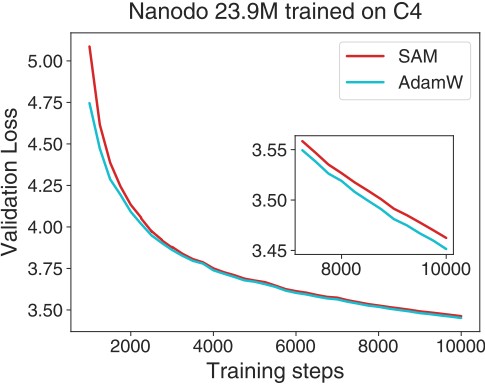

Figure 1: Evaluation loss curves of ADAMW and SAM for Nanodo decoder-only Transformer model (Liu et al., 2024) on the C4 dataset (Raffel et al., 2020).

*of the path* SAM *takes to sharpness reduction*, and show that it can be split into two contributions — one which modifies the *logit* statistics to reduce sharpness, and the other which modifies the geometry of the *function* itself. We measure these two contributions and find that in vision (where SAM works well) the contributions are relatively balanced; in contrast, in language modeling settings the logit path to sharpness minimization dominates.

We hypothesize that the functional path to sharpness minimization needs to be amplified in the language setting, and in **Section 4**, develop a *novel sharpness minimization algorithm* called FUNCTIONAL-SAM that achieves this. Additionally, we motivate another algorithm, PRECONDITIONED SAM, based on preconditioning, and give a theoretical argument that shows it promotes the functional path.

In **Section 5**, we show that FUNCTIONAL-SAM and PRECONDITIONED SAM provide improvements over baseline ADAMW when trained on the C4 dataset using the Nanodo Transformer codebase (Liu et al., 2024). Moreover, FUNCTIONAL-SAM and PRECONDITIONED SAM can be combined to yield maximal gains. This resulting combination consistently improves validation metrics in both fixed number of steps as well as Chinchilla-like scaling settings (Hoffmann et al., 2022) *at a variety of model sizes, spanning three orders of magnitude.*

We conclude with an extensive discussion, in **Section 8,** of additional SAM variants that may be useful in other domains as well as important future directions implied by our work.

## 2. Setup and Background

Let us assume that we are given data points $\mathbf{z} \in \mathcal{Z}$ drawn i.i.d. from some (unknown) distribution $\mathcal{D}$, where the samples $\mathbf{z}$ are input-output tuples $(\mathbf{x}, \mathbf{y})$, with the input $\mathbf{x} \in \mathbb{R}^d$ of dimension $d$ and the targets $\mathbf{y} \in \mathbb{R}^K$ of dimension $K$. We

seek to model the input-output relation via a neural network $\boldsymbol{f_\theta}(\mathbf{x}) : \mathbb{R}^d \mapsto \mathbb{R}^K$ with learnable parameters $\boldsymbol{\theta} \in \mathbb{R}^p$, such that $\boldsymbol{f_\theta}(\mathbf{x}) \approx \mathbf{y}$ , $\forall (\mathbf{x}, \mathbf{y}) \in \mathcal{D}$. We take the usual route of empirical risk minimization (Vapnik, 1991) and consider $\boldsymbol{\theta}^* := \operatorname{argmin}_{\boldsymbol{\theta} \in \boldsymbol{\Theta}} \mathcal{L}(\boldsymbol{\theta})$ with $\mathcal{L}(\boldsymbol{\theta}) := \frac{1}{n} \sum_{i=1}^n \boldsymbol{\ell}(\mathbf{z}_i; \boldsymbol{\theta})$ and where, $S = \{\mathbf{z}_i\}_{i=1}^n$ is the training set of size $n$ and $\boldsymbol{\ell}$ denotes the loss function. Hereafter we will consider the loss to be cross-entropy, which is the most popular choice; however, our analyses extend to other loss functions as well.

Under the above setup, Foret et al. (2020) formulates sharpness-aware minimization as the following min-max problem: $\min_{\boldsymbol{\theta}} \max_{\|\boldsymbol{\epsilon}\| \leq \rho} \mathcal{L}(\boldsymbol{\theta} + \boldsymbol{\epsilon})$ , where $\boldsymbol{\epsilon}$ denotes a perturbation of the parameters. In particular, the inner maximization is approximated to first order in the perturbation,

$$\max_{\|\boldsymbol{\epsilon}\| \leq \rho} \mathcal{L}(\boldsymbol{\theta}) + \boldsymbol{\epsilon}^\top \nabla_{\boldsymbol{\theta}} \mathcal{L}(\boldsymbol{\theta}) , \qquad (1)$$

which subsequently yields $\boldsymbol{\epsilon}^*(\boldsymbol{\theta}) = \rho \nabla_{\boldsymbol{\theta}} \mathcal{L}(\boldsymbol{\theta}) / \|\nabla_{\boldsymbol{\theta}} \mathcal{L}(\boldsymbol{\theta})\|$ as the optimal perturbation. In SAM (Foret et al., 2020), the authors propose making an update along the direction,

$$\mathbf{g}_{\text{SAM}} = -\nabla_{\boldsymbol{\theta}} \mathcal{L}(\boldsymbol{\theta} + \rho \boldsymbol{\epsilon}^*), \ \boldsymbol{\epsilon}^* \equiv \frac{\nabla_{\boldsymbol{\theta}} \mathcal{L}(\boldsymbol{\theta})}{\|\nabla_{\boldsymbol{\theta}} \mathcal{L}(\boldsymbol{\theta})\|} . \qquad (2)$$

## 3. The Dual Routes to Sharpness Minimization

In this section, we develop a diagnostic tool to understand the failures of SAM in language modeling settings. Our theoretical analysis shows that there are two possible routes to sharpness minimization via SAM— the *logit* path and the *functional* path. We derive quantities which can be used to measure the extent to which each route is active. We find that each path is relatively balanced in vision, but in language modeling settings (where SAM performs poorly) the logit path overwhelms the functional path.

### 3.1. The Penalty Formalization

We begin by looking at PENALTY-SAM, an alternative version of SAM that is often used in theoretical analyses (Andriushchenko & Flammarion, 2022; Dauphin et al., 2024) and whose gradient matches $\mathbf{g}_{\text{SAM}}$ to first order in $\rho$:

$$\min_{\boldsymbol{\theta}} \mathcal{L}(\boldsymbol{\theta}) + \underbrace{\rho \|\nabla_{\boldsymbol{\theta}} \mathcal{L}(\boldsymbol{\theta})\|}_{\text{SP}} , \qquad (3)$$

We can think of the added gradient norm term as a sharpness penalty SP. To understand how SP influences optimization, we investigate the structure of its gradient:

$$\nabla_{\boldsymbol{\theta}} \text{SP}(\boldsymbol{\theta}) := \rho \frac{\partial}{\partial \boldsymbol{\theta}} \|\nabla_{\boldsymbol{\theta}} \mathcal{L}(\boldsymbol{\theta})\| = \left( \frac{\partial}{\partial \boldsymbol{\theta}} \nabla_{\boldsymbol{\theta}} \mathcal{L}(\boldsymbol{\theta}) \right) \cdot \boldsymbol{\epsilon}^*(\boldsymbol{\theta}) . \qquad (4)$$

The gradient itself can be decomposed by the chain rule as

$$\nabla_{\boldsymbol{\theta}} \mathcal{L}(\boldsymbol{\theta}) := \nabla_{\boldsymbol{\theta}} \boldsymbol{F}(\boldsymbol{\theta}) \cdot \nabla_{\boldsymbol{F}} \mathcal{L}(\boldsymbol{\theta}) , \qquad (5)$$

where $\boldsymbol{F}(\boldsymbol{\theta}) = \left(\boldsymbol{f}_{\boldsymbol{\theta}}(\mathbf{x}_1)^\top \cdots \boldsymbol{f}_{\boldsymbol{\theta}}(\mathbf{x}_n)^\top\right)^\top \in \mathbb{R}^{Kn}$ collates the output over the entire dataset. The first term $\nabla_{\boldsymbol{\theta}} \boldsymbol{F}(\boldsymbol{\theta})$ represents the Jacobian (derivative of the model function outputs with respect to parameters); while the second term $\nabla_{\boldsymbol{F}} \mathcal{L}(\boldsymbol{\theta})$ is the derivative of the loss with respect to the outputs and which comes out to be the difference of the (softmax-ed) logits and the targets.

Using the product rule, we can rewrite Eqn. 4 as,

$$\underbrace{\left[\nabla_{\boldsymbol{\theta}} \boldsymbol{F}(\boldsymbol{\theta}) \cdot \frac{\partial}{\partial \boldsymbol{\theta}}\left(\nabla_{\boldsymbol{F}} \mathcal{L}(\boldsymbol{\theta})\right)\right] \cdot \boldsymbol{\epsilon}^*(\boldsymbol{\theta})}_{\text{SP gradient from logit perturbation} := \delta_{\texttt{logit}}} +$$

$$\underbrace{\left[\frac{\partial}{\partial \boldsymbol{\theta}}\left(\nabla_{\boldsymbol{\theta}} \boldsymbol{F}(\boldsymbol{\theta})\right) \cdot \nabla_{\boldsymbol{F}} \mathcal{L}(\boldsymbol{\theta})\right] \cdot \boldsymbol{\epsilon}^*(\boldsymbol{\theta})}_{\text{SP gradient from perturbing the function Jacobian} := \delta_{\texttt{func}}} . \quad (6)$$

The first term $\delta_{\texttt{logit}}$ represents the *logit-path* to sharpness minimization — directly optimizing the sharpness via the effect of the logits on the loss. In contrast, the second term $\delta_{\texttt{func}}$ represents the *functional-path* to sharpness minimization — that is, sharpness minimization via modification of the Jacobian statistics.

### 3.2. Contribution of Logit and Functional paths

We can explicitly relate the logit and functional paths to sharpness minimization ($\delta_{\texttt{logit}}$ and $\delta_{\texttt{func}}$ in Eqn. 6) with a decomposition of the Hessian of the loss $\mathbf{H}_{\mathcal{L}} = \nabla_{\boldsymbol{\theta}}^2 \mathcal{L}$,

$$\nabla_{\boldsymbol{\theta}} \mathbf{SP} = \delta_{\texttt{logit}} + \delta_{\texttt{func}} = \mathbf{H}_{\mathrm{G}} \cdot \boldsymbol{\epsilon}^* + \mathbf{H}_{\mathrm{F}} \cdot \boldsymbol{\epsilon}^* = \mathbf{H}_{\mathcal{L}} \cdot \boldsymbol{\epsilon}^* \quad (7)$$

where, we use the Gauss-Newton Decomposition of the Hessian (Schraudolph, 2002), namely:

$$\mathbf{H}_{\mathcal{L}} = \mathbf{H}_{\mathrm{G}} + \mathbf{H}_{\mathrm{F}} = \frac{1}{n}\sum_{i=1}^n \nabla_{\boldsymbol{\theta}} \boldsymbol{f}_{\boldsymbol{\theta}}(\mathbf{x}_i) \left[\nabla_f^2 \ell_i\right] \nabla_{\boldsymbol{\theta}} \boldsymbol{f}_{\boldsymbol{\theta}}(\mathbf{x}_i)^\top$$

$$+ \frac{1}{n}\sum_{i=1}^n \sum_{k=1}^K [\nabla_f \ell_i]_k \cdot \nabla_{\boldsymbol{\theta}}^2 \boldsymbol{f}_{\boldsymbol{\theta}}^k(\mathbf{x}_i) \quad (8)$$

where, the Generalized Gauss Newton (GGN) ($\mathbf{H}_{\mathrm{G}}$) and the functional Hessian ($\mathbf{H}_{\mathrm{F}}$) (Singh et al., 2021) are the component matrices of the loss Hessian ($\mathbf{H}_{\mathcal{L}}$). The GGN term captures the curvature of the linearized model; in contrast, the functional Hessian (also known as the Nonlinear Modeling Error) captures curvature due to model second derivatives (Dauphin et al., 2024).

Thus, we can interpret the *gradient through the logit and functional paths based on the Hessian component they depend upon* — the GGN and functional Hessian respectively.

**Normalized composition of Sharpness Gradient.** To better gauge which mode of sharpness gradient dominates, we will measure the following natural quantities,

$\tau_{\texttt{logit}}, \tau_{\texttt{func}}, \tau_{\texttt{cross}}$, where the three sum to 1:

$$\tau_{\texttt{logit}} := \frac{\|\delta_{\texttt{logit}}\|^2}{\|\nabla_{\boldsymbol{\theta}} \mathbf{SP}\|^2}, \tau_{\texttt{func}} := \frac{\|\delta_{\texttt{func}}\|^2}{\|\nabla_{\boldsymbol{\theta}} \mathbf{SP}\|^2},$$

$$\tau_{\texttt{cross}} := 2\frac{\langle \delta_{\texttt{logit}}, \delta_{\texttt{func}} \rangle}{\|\nabla_{\boldsymbol{\theta}} \mathbf{SP}\|^2} \quad (9)$$

Hence based on their values, *we can realize whether sharpness minimization will prioritise reduction of logit sharpness more or that of functional sharpness*, as well as how correlated they are by looking at the cross term.

### 3.3. Composition of Sharpness Gradient in Practice

We will now analyze SAM's behavior in language modeling, focusing on how the sharpness gradient composition reveals key differences between its application in language and vision tasks. More concretely, we consider next-token prediction task in the case of language using the C4 dataset and image classification in the case of vision. Since the typical vocabulary sizes in language is in the order of tens of thousands, so besides ImageNet-1K, we adopt other datasets like JFT (Sun et al., 2017) and ImageNet-21K (Ridnik et al., 2021) to make the settings further comparable in terms of number of outputs. For both settings, we employ Transformer-based networks, Nanodo (Liu et al., 2024), which is a simplified version of GPT-2 (Radford et al., 2019), in language modeling and Vision Transformer (ViT, Dosovitskiy et al., 2021) for vision tasks. Furthermore, in both cases, we train with ADAMW as the optimizer and measure the normalized sharpness gradient contributions (Eqn. 9) throughout training using exact Hessian-vector products. We present the results for Nanodo, C4 as well as ViT with ImageNet-1K and JFT in Figure 2, and that on ImageNet-21K in Appendix A.1.

**Observations.** Comparing these figures, we find a stark contrast between the language and vision settings. For vision, we find that the $\tau_{\texttt{logit}}$ starts close to 0 but that $\tau_{\texttt{logit}}$ and $\tau_{\texttt{func}}$ quickly become comparable for most of the training process (Figure 2, leftmost and second from left). In contrast for language, the logit-related gradient fraction $\tau_{\texttt{logit}}$ is close to 1 while the sharpness gradient related to the functional part is much smaller Figure 2 (second from right and rightmost). In all cases, $\tau_{\texttt{cross}}$ tends to be negative through most of training (see Figure 4 in the Appendix). This suggests that the two paths to sharpness regularization are antagonistic — taking one path moves you against the other. In language modeling, the dominance of the logit path, combined with negative $\tau_{\texttt{cross}}$, means that the overall contribution from the gradient of **SP** is unaligned, or even *anti-aligned*, with the functional path to sharpness.

**Logit Sharpness vs Functional Sharpness.** The above observations suggest that in NLP settings, SAM is heavily biased towards minimizing sharpness by reducing the

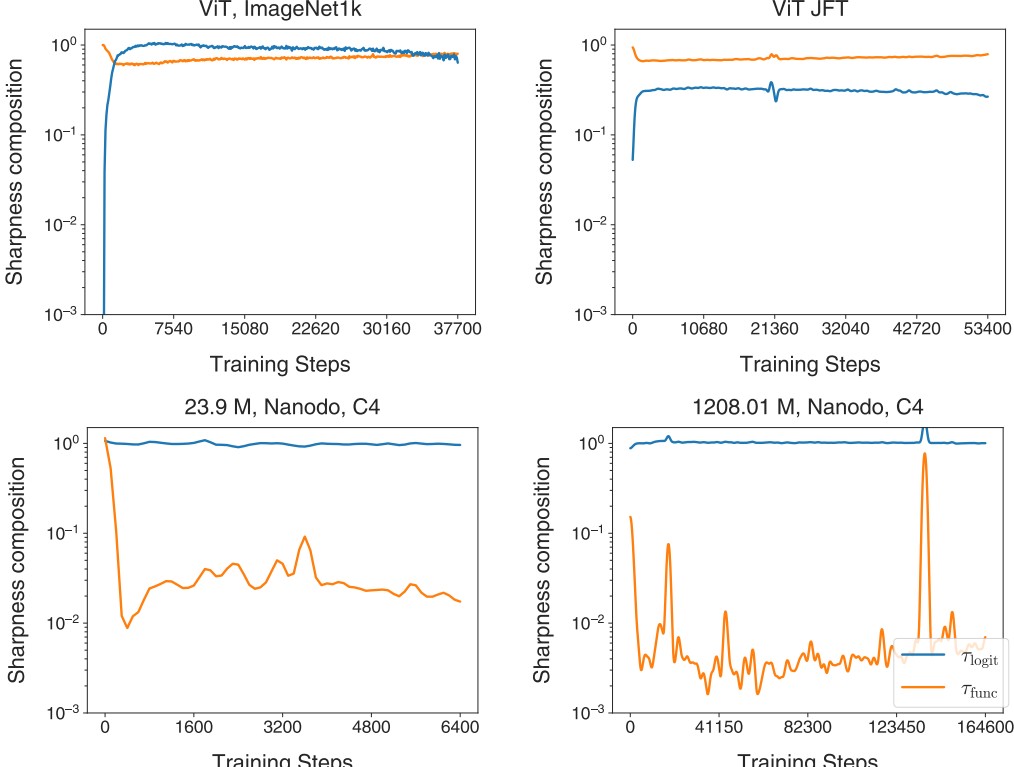

Figure 2: Normalized sharpness contributions $\tau_{\texttt{logit}}$ and $\tau_{\texttt{func}}$ show dramatically different trends across modalities. For ViT trained on ImageNet-1K (leftmost) and JFT (second from left), $\tau_{\texttt{logit}}$ starts near 0 but quickly increases to a comparable magnitude as $\tau_{\texttt{func}}$. For Transformer models trained on C4 (second from right and rightmost), $\tau_{\texttt{logit}} \gg \tau_{\texttt{func}}$ after the first few steps of training. This suggests that the pathways to sharpness regularization are more imbalanced in NLP compared to vision settings, which may contribute to the poor performance of SAM in NLP settings. $\tau_{\texttt{cross}}$ (plotted in Appendix A.1) is usually negative, suggesting the two methods of sharpness regularization tend to be antagonistic.

gradient of the loss with respect to the logits. There is not much sharpness reduction happening due to making the function more well-behaved. While both of these contribute to decreasing sharpness, the underlying mechanisms show interesting differences. Recalling the relation of logit-sharpness to the Gauss-Newton term $\mathbf{H}_{\mathrm{G}}$, we see that a simple but spurious way to decrease it is by making the network over-confident about its predictions. This is because of the presence of the term $\nabla_{\boldsymbol{f}}^2 \ell_i$, which equates to $\mathrm{diag}(\mathbf{p}_i) - \mathbf{p}_i \mathbf{p}_i^\top$, where $\mathbf{p}_i = \mathrm{softmax}(\boldsymbol{f_\theta}(\mathbf{x}_i))$. And thus as $\mathbf{p}_i$ becomes more one-hot (irrespective of leading to the correct output or the incorrect), the logit sharpness will get reduced.

In contrast, the functional sharpness is connected to the functional Hessian, and the ways of decreasing it (such as via modeling the target better $\mathbf{p}_i \to \mathbf{y}_i$; or reducing the second-derivative of the function with respect to the parameters $\|\nabla_{\boldsymbol{\theta}}^2 \boldsymbol{f}\| \to 0$ and hence encouraging a functional simplicity) are intuitively more desirable, even though setting a particular proportion of it might be unclear.

These observations lead to the *hypothesis that if we could encourage a reduction of functional sharpness in lieu of logit sharpness, then sharpness minimization might in fact work for language modeling too.* In the next section, we operate under this hypothesis and derive simple but principled modifications of the SAM update rule.

## 4. Algorithms to promote the functional path to sharpness minimization

From the PENALTY-SAM formulation in Eqn. 3 and the form of the corresponding sharpness gradients in Eqn. 7, we can see that the functional path to sharpness can, in principle, be amplified by manipulating $\delta_{\texttt{logit}}$ and $\delta_{\texttt{func}}$. However, while PENALTY-SAM is competitive with the SAM algorithm, it is less robust than SAM due to ill-behaved second derivatives (Dauphin et al., 2024). In contrast to PENALTY-SAM, the perturbation approach in SAM does not explicitly involve second-derivatives in the gradient. Therefore, it is important to develop strategies that promote

the functional path using the methodology taken in SAM.

In this section, we discuss two strategies — one direct and the other indirect — that promote the functional path to sharpness minimization, using algorithms which maintain the benefits of the SAM formulation.

### 4.1. FUNCTIONAL-SAM

The simplest way to promote the functional path would be to use a perturbation which aligns more with $\delta_{\texttt{func}}$ than $\delta_{\texttt{logit}}$. We can find such a perturbation by decomposing the SAM update rule from Eqn. 2, as follows:

$$\nabla_{\boldsymbol{\theta}}\mathcal{L}(\boldsymbol{\theta}+\rho\boldsymbol{\epsilon}^*) = \underbrace{[\nabla_{\boldsymbol{\theta}}\boldsymbol{F}(\boldsymbol{\theta})\cdot\nabla_{\boldsymbol{F}}\mathcal{L}(\boldsymbol{\theta}+\rho\boldsymbol{\epsilon}^*) - \nabla_{\boldsymbol{\theta}}\mathcal{L}(\boldsymbol{\theta})]}_{\delta_{\texttt{logit}}\text{ up to first order in }\rho}$$

$$+ \underbrace{[\nabla_{\boldsymbol{\theta}}\boldsymbol{F}(\boldsymbol{\theta}+\rho\boldsymbol{\epsilon}^*)\cdot\nabla_{\boldsymbol{F}}\mathcal{L}(\boldsymbol{\theta}) - \nabla_{\boldsymbol{\theta}}\mathcal{L}(\boldsymbol{\theta})]}_{\delta_{\texttt{func}}\text{ up to first order in }\rho} + \nabla_{\boldsymbol{\theta}}\mathcal{L}(\boldsymbol{\theta}) + O(\rho^2)$$

$$(10)$$

The first term is $\delta_{\texttt{logit}}$ up to first order in $\rho$, and the second term is $\delta_{\texttt{func}}$ up to first order in $\rho$. This suggests the following update rule which we call FUNCTIONAL-SAM:

$$\mathbf{g}_{\text{FUNC-SAM}} = -\nabla_{\boldsymbol{\theta}}\boldsymbol{F}(\boldsymbol{\theta}+\rho\boldsymbol{\epsilon}^*)\cdot\nabla_{\boldsymbol{F}}\mathcal{L}(\boldsymbol{\theta}) \qquad (11)$$

where, we discard the $\delta_{\texttt{logit}}$ contribution. Further, FUNCTIONAL-SAM uses the same perturbation $\boldsymbol{\epsilon}^*$ as the SAM formulation, but *only* perturbs the Jacobian — thus emphasizing the functional path to sharpness as desired. This update rule can be implemented as efficiently as regular SAM using the very same vector-Jacobian product operations that are used to compute gradient in most autodifferentiation frameworks. In Appendix C, we provide the JAX code snippets for SAM and FUNCTIONAL-SAM, demonstrating that the difference in their implementation is a matter of a few lines. Further, like SAM, FUNCTIONAL-SAM remains compatible with methods like ADAMW which take a gradient and then further process it.

Overall, the update rule in Eqn. 11 has the same cost as the SAM update rule, and keeps the benefit of the finite-differences based perturbation approach to sharpness estimation that keeps SAM robust — meeting all our initial goals.

### 4.2. PRECONDITIONED SAM

In language modeling, Transformers (Vaswani, 2017) are almost exclusively trained with ADAMW or other adaptive methods, and SGD-based training is known to be significantly worse (Liu et al., 2020). This is commonly attributed to the presence of heterogeneity in the gradients (Liu et al., 2020; Noci et al., 2022; Pan & Li, 2023) and the curvature (Zhang et al., 2024; Ormaniec et al., 2024; Jiang et al., 2024) across different "modules" (layer types, layers at different depths, etc.), with the idea that ADAMW alleviates

heterogeneity and improves conditioning.

Naively combining SAM and ADAMW creates a potential mismatch — the SAM perturbation is carried out with respect to the unpreconditioned geometry. We hypothesize that this mismatch offers grounds for spurious sharpness minimization. To rectify this, we consider preconditioning the perturbation $\boldsymbol{\epsilon}^*$ with the inverse of the same second-moment statistics $\mathbf{M}$ from ADAMW, resulting in the update:

$$\mathbf{g}_{\text{PRECOND SAM}} \equiv -\nabla_{\boldsymbol{\theta}}\mathcal{L}\left(\boldsymbol{\theta} + \widetilde{\rho}\,\mathbf{M}^{-1}(\boldsymbol{\theta})\,\nabla_{\boldsymbol{\theta}}\mathcal{L}(\boldsymbol{\theta})\right), \quad (12)$$

where $\widetilde{\rho} := \rho/\|\mathbf{M}^{-1}(\boldsymbol{\theta})\,\nabla_{\boldsymbol{\theta}}\mathcal{L}(\boldsymbol{\theta})\|$, and $\mathbf{g}_{\text{PRECOND SAM}}$ is then passed to the rest of Adam in lieu of the gradient $\nabla_{\boldsymbol{\theta}}\mathcal{L}(\boldsymbol{\theta})$.

Another motivation for PRECONDITIONED SAM is that the gradients often align with the principal eigenspaces of $\mathbf{H}_{\text{G}}$ (Gur-Ari et al., 2018). This would amplify $\delta_{\texttt{logit}}$ over $\delta_{\texttt{func}}$, since $\delta_{\texttt{logit}} = \mathbf{H}_{\text{G}} \cdot \boldsymbol{\epsilon}^*$ and $\boldsymbol{\epsilon}^*$ is parallel to the gradient. Preconditioning $\boldsymbol{\epsilon}^*$ by $\mathbf{H}_{\text{G}}^{-1}$ reduces this effect, thereby *promoting the functional path over the logit path* (see Appendix B.1 for a more detailed argument). Although $\mathbf{H}_{\text{G}}^{-1}$ would be the best preconditioner in this regard, it is expensive to estimate generally and ADAMW already gives us a diagonal estimator in its own preconditioner $\mathbf{M}^{-1}$ at no additional cost.

To conclude, this algorithm (Eqn. 12) has only marginally higher computational cost than standard SAM + ADAMW, and provides an indirect way to improve the contribution of the functional path to sharpness minimization.

## 5. Empirical Evaluation

### 5.1. Setup

**Training Lengths**. In order to evaluate language models of multiple sizes, we consider (pre-)training them in two scenarios: (a) when the training length is kept fixed across scales and (b) when the training length is adjusted as per compute-optimality considerations (Kaplan et al., 2020; Hoffmann et al., 2022).

*(a) Fixed-Length Training Scheme.* In this setup, we train all the models for 10K steps, which amounts to seeing roughly a total of 1.3 billion tokens. The results for this setting are presented in Table 2. *(b) Chinchilla-style Training Horizon Scheme.* Unlike traditional image-classification based scenarios, training language models often involves determining compute-optimal scaling laws, and whereby models are trained on a corpus size in proportion to their parameters. Specifically, we follow the $20\times$ over-training policy suggested in the Chinchilla (Hoffmann et al., 2022) training regime, and use it together with the fixed-depth scheme, as considered in Everett et al. (2024). The corresponding results are shown in Table 3.

**Training Details and Hyperparameters**. In both scenarios, we consider a batch size of 256 sequences, of maximum length 512, and evaluate model at 5 different sizes: 2M (for prototyping), 23.9M, 42.5M, 117.9M, and 1208M in terms of non-embedding parameters (see details in Appendix A.6), and trained with ADAMW (Kingma & Ba, 2017; Loshchilov & Hutter, 2019) as the underlying optimizer on the C4 dataset (Raffel et al., 2020). We used a decoupled weight decay parameter set to $10^{-4}$ for all experimental settings. We use the Nanodo (Liu et al., 2024) framework to implement these minimal decoder-only Transformer models, in Flax (Heek et al., 2024) together with JAX (Bradbury et al., 2018).

### 5.2. Comparison of direct and indirect approaches

Before carrying out an extensive evaluation across different model scales, we do initial prototyping on a smaller model size. This lets us save computational resources and scale up the most promising methods. In particular, we are interested in knowing which methods out of direct and indirect approaches, and their combinations, are most relevant. In Table 1, we present results on a model with 2M non-embedding parameters trained in the fixed-length regime.

Table 1: *Effect of Preconditioning the SAM perturbation on* FUNC-SAM *with* 2M *parameter model when trained for* 50K *steps and at a peak learning rate of* 0.001. *Lower is better.*

| METHOD | EVAL LOSS |
|---|---|
| ADAMW | 3.901 |
| SAM | 3.910 |
| PRECOND SAM | 3.889 |
| FUNC-SAM | 3.880 |
| PRECOND FUNC-SAM | **3.861** |

We notice that the plain version of FUNCTIONAL-SAM outperforms both ADAMW and preconditioned SAM. But, FUNCTIONAL-SAM can be further improved by using preconditioning alongside, yielding a significant improvement in terms of evaluation loss. Also, it should be noted that the standard deviation across different seeds is typically around $\sim 0.002$. Moreover, mid-third decimal differences in evaluation loss are typical of the gains provided by new optimization methods for pre-training LLMs look like, e.g., in CASPR (Duvvuri et al., 2024), and even in newer variants of attention (Leviathan et al., 2024), as well as when fusing LLMs (Mavromatis et al., 2024) or when deduplicating training corpus (Lee et al., 2021). Thus, the kind of gains shown in Table 1 are indeed significant.

The experiments suggest that FUNCTIONAL-SAM also ben-

efits from preconditioning; we hypothesize that the functional route to sharpness also benefits from reduction in heterogeneity across modules and better matching between inner and outer optimization geometries. Hereafter, we will use the preconditioned variant for FUNCTIONAL-SAM, instead of just the plain FUNCTIONAL-SAM, to reduce the computational costs associated with testing at larger model scales. This relationship merits further study, which is outside the scope of our work.

### 5.3. Results at Multiple Model Scales

**Hyperparameter Setup.** For thoroughness, *at each model size separately*, we simultaneously tune learning rate and perturbation strength $\rho$ in our experiments, even though this may be difficult at even bigger model sizes or when faced with computational constraints. The upside of this much harder testing ground is to provide us the key assurance that the resulting gains cannot be obtained through other means such as strong hyperparameter tuning of baselines. These results can be found in Tables 2 and 3.[2]

In the case of billion-plus parameter model, this procedure however amounted to extremely high computational costs, and so we instead tuned the learning rate for the baseline AdamW first. Subsequently, we tuned $\rho$ for all SAM methods based on this corresponding optimal learning rate for AdamW.

**Observations.** We find that in both the training regimes, our proposed algorithms, namely PRECOND FUNCTIONAL-SAM and PRECOND SAM, significantly outperform SAM, at all the model scales in Tables 2 and 3. The gains typically range from $5 \cdot 10^{-3} - 10^{-2}$, are largest for 117.9M parameters. We find that SAM performs the worst of the lot, even worse than ADAMW while being $2\times$ computationally expensive. The numbers reported for SAM represent the smallest $\rho$ tested (0.1), since it monotonically gets worse with increasing $\rho$. Overall, we find that PRECOND FUNCTIONAL-SAM achieves the best results, followed by a mix of PRECOND SAM and ADAMW.

At the billion-parameter scale, PRECOND SAM and SAM show susceptibility to numerical instabilities (yielding `NaNs`) across training regimes, whereas PRECOND FUNCTIONAL-SAM *is significantly more robust*. In the relatively cheaper fixed-length runs where we could carry out additional investigation, we find that warming up the perturbation radius or a lower learning rate choice can sometimes mitigate these issues for PRECOND SAM and SAM, however, they are still outperformed by our best

---

[2]For brevity, we omit the standard deviation for the baselines in these Tables. But in the Appendix, the reader can also find the full table with standard deviations for all baseline methods as well. The standard deviation for the baselines results are nevertheless very similar.

Table 2: Evaluation loss comparison of different methods in a *fixed-length (10K steps) training setup*, where the optimal learning rate is found for each model size separately. *Results are averaged over 5 seeds.* For completeness, the optimal values of the perturbation strength for PRECOND FUNCTIONAL-SAM at the various scales are respectively $0.5, 0.5, 0.4, 0.4$. Lower is better.

| SIZE | PRECOND FUNC-SAM | PRECOND SAM | SAM | ADAMW |
|---|---|---|---|---|
| 23.9 M | $\mathbf{3.425}_{\pm 0.5 \times 10^{-3}}$ | 3.431 | 3.450 | 3.430 |
| 42.5 M | $\mathbf{3.344}_{\pm 2.2 \times 10^{-3}}$ | 3.350 | 3.366 | 3.349 |
| 117.9 M | $\mathbf{3.218}_{\pm 2.2 \times 10^{-3}}$ | 3.224 | 3.257 | 3.228 |
| 1208 M | $\mathbf{3.048}_{\pm 1.5 \times 10^{-3}}$ | 3.059 | NaN | 3.056 |

method, PRECOND FUNCTIONAL-SAM, which obtains a lower evaluation loss and does not need a perturbation warm-up in either training regime.

Table 3: Evaluation loss comparison of different methods in *Chinchilla like training (Everett et al., 2024)*, where the optimal learning rate is found for each model size separately. *All results have been averaged over 5 seeds*, except for the more expensive billion parameter runs which are based on a single seed. For completeness, the optimal values of the perturbation strength for PRECOND FUNCTIONAL-SAM at the various scales are respectively $0.7, 0.5, 0.4, 0.5$. Lower is better.

| SIZE | PRECOND FUNC-SAM | PRECOND SAM | SAM | ADAMW |
|---|---|---|---|---|
| 23.9 M | $\mathbf{3.492}_{\pm 2.9 \times 10^{-3}}$ | 3.498 | 3.516 | 3.497 |
| 42.5 M | $\mathbf{3.340}_{\pm 2.0 \times 10^{-3}}$ | 3.348 | 3.359 | 3.344 |
| 117.9 M | $\mathbf{3.070}_{\pm 2.2 \times 10^{-3}}$ | 3.074 | 3.094 | 3.079 |
| 1208 M | $\mathbf{2.557}$ | 2.562 | NaN | 2.561 |

We note that for the 1208M parameter models, $\rho$ was tuned more coarsely due to computational costs. Despite less tuning, the gains for PRECOND FUNCTIONAL-SAM persist at this scale in the Chinchilla training horizon experiments.

In Tables 6 and 7 of the appendix, we also illustrate how the results fare if a common learning rate of 0.0001 is used across all model scales. Interestingly, we find that PRECOND FUNCTIONAL-SAM is much less sensitive to the lack of rightly tuned learning rate than the baselines. As a matter of fact, in such settings we find the corresponding gains to be even bigger.

### 5.4. Discussion

The above-mentioned results demonstrate that the issue with SAM in NLP can be successfully resolved through PRECOND FUNCTIONAL-SAM. The significant improvements in validation performance above are *especially intriguing* if we bear in mind that these are obtained (a) even when using a *clean corpus such as C4* and (b) training in an *online fashion where no batch is seen more than once* due to massive size of the C4 corpus (even when training for longer duration like in the Chinchilla setup). These are ideal conditions which may not always hold. This suggests that further improvements in generalization may occur when training on noisy corpora or in multipass settings; we leave exploration of these potential effects for future work.

All in all, these results confirm the benefits imparted by FUNCTIONAL-SAM and PRECONDITIONED SAM over SAM, and these improvements in generalization over ADAMW *restore the promise of sharpness regularization.*

## 6. Ablation Studies

**Perturbation strengths.** In the above-mentioned results, the best values for SAM typically occur around perturbation radius $\rho = 0.1$, which, in our experiments, happens to be the smallest nonzero perturbation radius considered. However, if we employ larger perturbation radii, the performance of SAM rapidly deteriorates, as shown in Figure 3 (left), which moreover suggests that the optimal value of perturbation size $\rho$ is 0 — i.e., *not using* SAM *at all*. In contrast, the optimal value of perturbation for PRECOND FUNCTIONAL-SAM tends to be much larger, across all values of learning rate as shown in Figure 3 (right). This explains why we needed to modify the relative magnitudes of the sharpness contributions using PRECOND FUNCTIONAL-SAM— the logit term degrades performance at even small $\rho$, overwhelming the potential gains from the functional path to sharpness minimization.

**Non-linearity Choice.** While all the above experimental results utilize GeLU non-linearity, we also carry out experiments with ReLU as the choice of non-linearity. These results can be found in Table 8 of the Appendix, but the key observation is that similar improvements are observed in the evaluation loss when using PRECOND FUNCTIONAL-SAM for ReLU based architectures as well.

### 6.1. Sharpness of Final Solutions

Here, we confirm that the generalization benefits imparted by PRECOND FUNCTIONAL-SAM are brought about convergence to a solution with lower curvature, as shown in the Table 4 for the 23.9M model. Similar results can also be found on other model sizes, and Table 9 of the Appendix shows them for the 117.9M model.

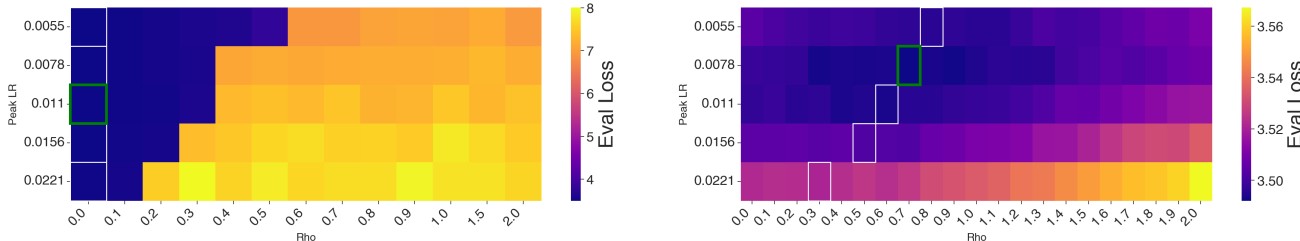

Figure 3: *Effect of increasing perturbation strength $\rho$ for* SAM *(left) and* PRECOND FUNCTIONAL-SAM *(right) across various peak learning rates, in an equal compute setup.* Each cell shows the evaluation loss, averaged over 5 seeds, at the corresponding values of learning rate and perturbation radius. The best performing value of perturbation strength at a given learning rate is marked by white squares, while the overall best learning rate and perturbation strength pair is marked in green. We see that with SAM any non-zero $\rho$ does worse than the baseline ($\rho = 0$), while PRECOND FUNCTIONAL-SAM, with the same compute costs, shows improvements at all values of peak learning rates (Nanodo trained on C4, 23.9M parameters, Chinchilla like setup).

Table 4: Comparison of different methods based on Hessian $\mathbf{H}_{\mathcal{L}}$ and GGN $\mathbf{H}_{\mathrm{G}}$ maximum eigenvalue and trace for the 23.9M model trained as per Chinchilla like training setup, with a peak learning rate of 0.001. Lower is better for all metrics. The best entry is in **bold**, the second best is underlined.

| METHOD | EVAL LOSS | $\lambda_{\max}(\mathbf{H}_{\mathcal{L}})$ | $\mathrm{tr}(\mathbf{H}_{\mathcal{L}})$ | $\mathrm{tr}(\mathbf{H}_{\mathrm{G}})$ |
|---|---|---|---|---|
| ADAMW | 3.688 | 10.61 | 4897.52 | 4745.03 |
| SAM | 3.706 | **2.71** | 3324.58 | 3231.46 |
| PRECOND SAM | 3.663 | 5.62 | 3182.78 | 3097.87 |
| PRECOND FUNC-SAM | **3.631** | 6.20 | **2687.12** | **2503.86** |

We notice that all sharpness minimization methods yield lower value of the maximum eigenvalue and the trace, as compared to ADAMW. Interestingly, we also find that SAM results in the lowest Hessian maximum eigenvalue, even though it performs much worse than ADAMW when evaluating on the validation set. This further highlights how SAM, by default, in language modeling tasks is set up to minimize sharpness spuriously. In contrast, we see that for both of our proposed methods, the improved generalized performance comes hand-in-hand with better landscape properties of their solutions.

## 7. Related work

**Improved variants of SAM in Vision.** An extensive line of work has attempted to propose better definitions of sharpness — particularly those which are less sensitive to details of parameterization (Kwon et al., 2021; Tahmasebi et al.,

2024; Li & Giannakis, 2024). Some of these methods have shown small improvements on vision tasks. *We believe our decomposition approach is orthogonal to this line of research.* Therefore, the obtained FUNCTIONAL-SAM algorithm is substantially different from prior work.

**Studies exploring preconditioning for SAM.** Other work has also suggested that the perturbation step in SAM should be taken in an alternative geometry. Our approach to preconditioning is most similar to Fisher SAM (Kim et al., 2022), and the concurrent work of Zhang et al. (2025) which describes a more general preconditioning scheme for SAM. Our key insight is that it is useful to take the SAM perturbation in the *exact same geometry* used by the optimizer, which can be accomplished for negligible cost in the case of Adam and its variants. Furthermore, our work is *primarily driven by the problem of making* SAM *work in language modeling*, which is far from the focus of these other works.

**Role of the indefinite Hessian term.** At a more conceptual level, our study aligns with recent works (Singh et al., 2021; Dauphin et al., 2024) which underscore paying more importance to the functional Hessian, the understudied indefinite term of the Hessian in the Gauss-Newton decomposition, as opposed to focusing solely on the positive semi-definite GGN term as suggested by prior studies (Sagun et al., 2018; Papyan, 2019; Jacot et al., 2020). The decomposition of the sharpness gradient into logit and functional modes, along with the demonstrated significance of FUNCTIONAL-SAM in NLP tasks (which is closely tied to the functional Hessian), *highlights the risks of over-reliance on the GGN and the corresponding outlier spectrum of the Hessian* — especially in the context of *regularization* of sharpness.

**SAM in NLP.** Prior works building on SAM in NLP have been restricted to the fine-tuning setting (Bahri et al., 2022), domain transfer (Sherborne et al., 2024) or on small-scale machine translation setups (Li & Giannakis, 2024). To

the best of our knowledge, SAM has hitherto not been successfully applied to language modeling, particularly in any scaling setting.

## 8. Discussion and Future Work

There are several important aspects that we would like to elaborate on, which could spark interesting future work:

**Interpolating smoothly between paths to sharpness minimization.** In this work, we focused on FUNCTIONAL-SAM due to our diagnosis of issues in language modeling. However, one can imagine that there might be other domains where FUNCTIONAL-SAM gives spurious minimization, and the alternative "LOGIT-SAM" may need to be emphasized. Our implementation of FUNCTIONAL-SAM can be extended to a more continuous "ANGLE-SAM" which can smoothly interpolate between the extremes, which we discuss in detail in Appendix D. Though we did not find ANGLE-SAM necessary in the language modeling setting, there might be other model-dataset-optimizer triples where it would prove beneficial.

**The Perturbation Scope.** One of the key takeaways from this work is how scoping the perturbation to the level of the function Jacobian suffices to enable the benefits of SAM in NLP tasks. This scoping can be generalized to *any* set of paths or branches in the computational graph. For example, the output of network with residual connections can be decomposed into multiple streams like $f(\mathbf{x}) = f_1(\mathbf{x}) + f_2(\mathbf{x}) + \cdots + f_m(\mathbf{x})$ and practitioners can choose to either have the perturbation go through all or some of them. This perspective opens the door to many more creative levels of perturbation scoping and new regularization techniques.

**Optimal Perturbation Transfer.** At the largest scales, extensive tuning of the perturbation radius is unfeasible; FUNCTIONAL-SAM and its variants will benefit from improved parameterizations that optimally transfer the perturbation radius across model scales. The recently explored layerwise perturbation scaling regime for SAM in vision tasks (Haas et al., 2024) may be promising for language too.

**Utility for Downstream tasks and deployment.** An added advantage of training with sharpness minimization methods is that the resulting flatter solutions can adapt more gracefully to the post-training pipeline, like downstream fine-tuning or model compression. These benefits have been demonstrated in prior work such as (Liu et al., 2023), where a lower Hessian trace at the solution has been shown to correlate better with performance on downstream tasks, and we expect similar benefits to also hold with FUNCTIONAL-SAM and its variants, given the obtained flatter solutions (see Tables 4, 9). Likewise prior work (Na et al., 2022) has also advocated the use of such methods when subsequent model compression is intended, and we also present a very

simple demonstration with one-shot pruning in Figure 5 where we see that FUNCTIONAL-SAM shows a more graceful degradation as opposed to ADAMW with increasing sparsity. We leave a detailed study to future work.

**Efficiency.** A common drawback of sharpness minimization methods is that they require twice the gradient computation per step, and hence are twice as expensive as compared to other optimizers like ADAMW. In this paper, our singular focus has been to address the ineffectiveness of SAM in language modeling, which we have been able to carry out successfully through the proposed FUNCTIONAL-SAM and its preconditioning variants. However, these algorithms still share the same $2\times$ computational burden like SAM itself, and thus additional work is required in the future to make it suitable for deployment. Thankfully, since FUNCTIONAL-SAM is structured similarly to SAM, most advancements in efficient implementations of SAM should be useful for FUNCTIONAL-SAM as well. Given the plethora of recent work in this area for vision (Du et al., 2022; Liu et al., 2022; Becker et al., 2024; Xie et al., 2024), we are optimistic about the development of efficient FUNCTIONAL-SAM variants for language modeling in the near future.

**Data-bound scenarios and model size constraints.** In any case, we would like to emphasize that our primary comparison point in the paper is that at equal number of steps — which is becoming an increasingly relevant scenario. This is because industrial settings are often bound by data availability or a limited model size (e.g., inference constraints). Therefore, here extra training time is acceptable for better final quality.

## 9. Conclusion

Our work thoroughly highlights the ineffectiveness of SAM for language modeling and uncovers the underlying reasons behind such an occurrence. We show that this arises because sharpness reduction in NLP settings is prone to logit manipulation, and hence spurious sharpness minimization, — rather than being driven by promoting the simplicity of the network function. Based on this insight, we propose addressing this issue via FUNCTIONAL-SAM and preconditioning. We demonstrate that both these simple but principled modifications to SAM restore its generalization properties across multiple model scales.

More broadly, we believe that our novel foray into functional and logit modes of sharpness reduction will reinvigorate the existing research into SAM, and pave the way for advanced curvature regularization techniques. Lastly, we are excited about the nuanced characterization of sharpness introduced here and hope that it advances our fundamental understanding of sharpness, its dynamics, and the broader nature of loss landscapes.

## Impact Statement

This paper presents work whose goal is to advance the field of Machine Learning. There are many potential societal consequences of our work, none which we feel must be specifically highlighted here.

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

# A. Additional Results

## A.1. Sharpness Composition Plots

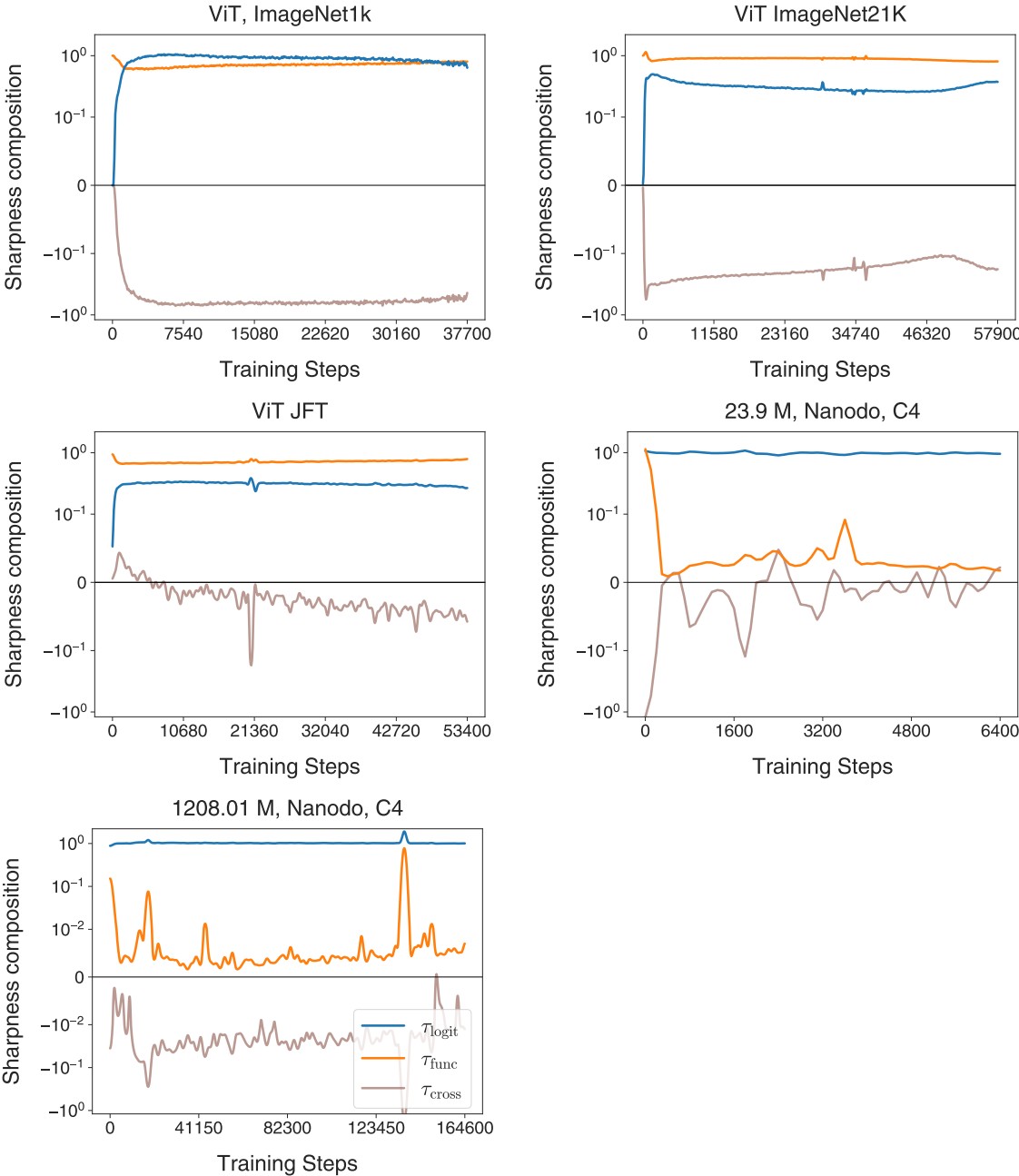

Figure 4: Sharpness contributions $\tau_{\mathtt{logit}}$, $\tau_{\mathtt{func}}$ and $\tau_{\mathtt{cross}}$ for various datasets. $\tau_{\mathtt{cross}}$ tends to be negative for most of training.

## A.2. Detailed Table Results

Below in Table 5, we include the standard deviation for each of the baseline method as well, in addition to that listed for PRECOND FUNCTIONAL-SAM in Table 3.

Table 5: Evaluation loss comparison of different methods in *Chinchilla like training (Everett et al., 2024)*, where the optimal learning rate is found for each model size separately. All results have been averaged over 5 seeds, except for the more expensive billion parameter runs which are based on a single seed. Lower is better.

| SIZE | PRECOND FUNC-SAM | PRECOND SAM | SAM | ADAMW |
|------|------------------|-------------|-----|-------|
| 23.9 M | **3.492**$_{\pm 2.9 \times 10^{-3}}$ | 3.498$_{\pm 3.6 \times 10^{-3}}$ | 3.516$_{\pm 2.9 \times 10^{-3}}$ | 3.497$_{\pm 2.9 \times 10^{-3}}$ |
| 42.5 M | **3.340**$_{\pm 2.0 \times 10^{-3}}$ | 3.348$_{\pm 1.7 \times 10^{-3}}$ | 3.359$_{\pm 3.4 \times 10^{-3}}$ | 3.344$_{\pm 2.5 \times 10^{-3}}$ |
| 117.9 M | **3.070**$_{\pm 2.2 \times 10^{-3}}$ | 3.074$_{\pm 4.0 \times 10^{-3}}$ | 3.094$_{\pm 3.1 \times 10^{-3}}$ | 3.079$_{\pm 2.7 \times 10^{-3}}$ |
| 1208 M | **2.557** | 2.562 | NaN | 2.561 |

## A.3. Results across Scales at a common learning rate

Table 6: Evaluation loss comparison of different methods in a *fixed-length (10K steps) training setup* at a common learning rate of 0.001. Lower is better.

| SIZE | PRECOND FUNC-SAM | PRECOND SAM | SAM | ADAMW |
|------|------------------|-------------|-----|-------|
| 23.9 M | **3.527** | 3.547 | 3.587 | 3.565 |
| 42.5 M | **3.414** | 3.439 | 3.462 | 3.451 |
| 117.9 M | **3.252** | 3.266 | 3.288 | 3.280 |
| 1208 M | **3.054** | NaN | NaN | 3.081 |

Table 7: Evaluation loss comparison of different methods in *Chinchilla like training (Everett et al., 2024)* at a common learning rate of 0.001. Lower is better.

| SIZE | PRECOND FUNC-SAM | PRECOND SAM | SAM | ADAMW |
|------|------------------|-------------|-----|-------|
| 23.9 M | **3.631** | 3.663 | 3.706 | 3.688 |
| 42.5 M | **3.414** | 3.435 | 3.460 | 3.445 |
| 117.9 M | **3.096** | 3.108 | 3.126 | 3.120 |
| 1208 M | **2.612** | NaN | NaN | 2.627 |

### A.4. Results across non-linearities

Table 8: Comparison of SAM variants across non-linearities for a Nanodo model with 2M (non-embedding parameters) on C4 dataset.

| NON-LINEARITY | METHOD | EVAL LOSS |
|---|---|---|
| GeLU | PRECOND FUNCTIONAL-SAM | 3.8614 |
| GeLU | PRECOND SAM | 3.8894 |
| GeLU | ADAMW | 3.9069 |
| ReLU | PRECOND FUNCTIONAL-SAM | 3.8777 |
| ReLU | PRECOND SAM | 3.8937 |
| ReLU | ADAMW | 3.9145 |

### A.5. Hessian spectra results at another model size

Table 9: Comparison of different methods based on Hessian $\mathbf{H}_{\mathcal{L}}$ and GGN $\mathbf{H}_{\mathrm{G}}$ metrics for the 117.9M model trained as per Chinchilla like training setup.

| METHOD | EVAL LOSS | $\lambda_{\max}(\mathbf{H}_{\mathcal{L}})$ | $\mathrm{tr}(\mathbf{H}_{\mathcal{L}})$ | $\mathrm{tr}(\mathbf{H}_{\mathrm{G}})$ |
|---|---|---|---|---|
| PRECOND FUNC-SAM | **3.096** | 8.884 | *2790.650* | 2746.559 |
| PRECOND SAM | 3.108 | *7.415* | 2920.445 | *2060.214* |
| SAM | 3.126 | **3.381** | **2235.759** | **2222.779** |
| ADAMW | 3.120 | 9.259 | 3299.497 | 3285.881 |

### A.6. Architectural Details.

The 23.9M, 42.5M, 117.9M, and 1208M models have the same depth of 6, and whose width has been scaled together with the number of heads. In particular, these correspond to $h = 9, 12, 20,$ and 64 heads per block and the width $m$ scales as $m = 64 \times h$, and the MLP dimension is $f = 4 \times m$. The 2M model used for prototyping has depth 3, 4 heads per block, width $m = 256$ and MLP dimension $f = 1024$.

### A.7. Downstream benefits of sharpness minimization

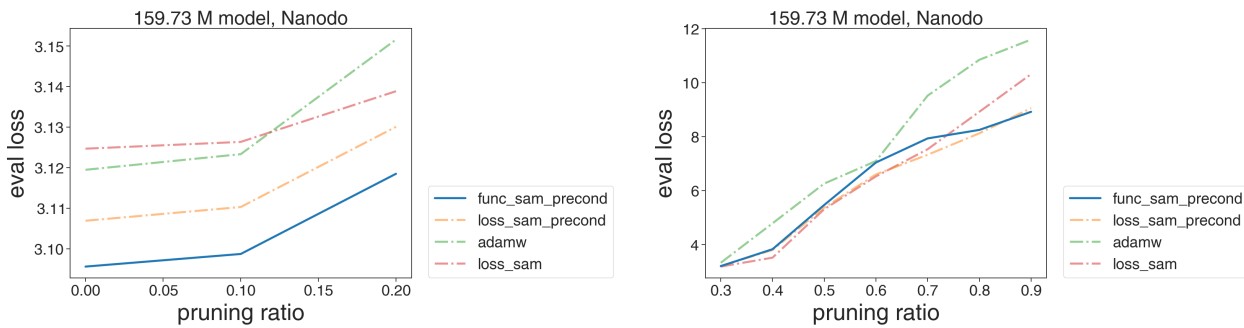

Figure 5: *Effect of one-shot (unstructured) global magnitude pruning:* We see that sharpness minimization methods tend to degrade more gracefully as increasing number of parameters are pruned. Also, from this figure we can see that the performance gained imparted by FUNCTIONAL-SAM over ADAMW is equivalent to setting about $25\%$ parameters of zero, and is thus significant.

## B. Additional theory

### B.1. Argument for reducing matrix-vector products with inverse preconditioning

We will use a random matrix model to reason about the effects of preconditioning with a matrix inverse. Random matrices have been used to model the spectra of the Hessian of the large models found in machine learning, and in particular treating the Gauss-Newton and the functional Hessian/NME as independent yields quantitative insights about the structure of the overall Hessian (Pennington & Bahri, 2017).

Consider two $N \times N$ symmetric invertible matrices $\mathbf{A}$ and $\mathbf{B}$. Suppose $\mathbf{A}$ and $\mathbf{B}$ are random and freely independent. Free independence is the non-commutative analog to classical independence, implied by classical independence of entries in the limit of large matrix size (Pennington & Bahri, 2017). The key feature it induces is

$$\mathbb{E}[\text{tr}[\mathbf{A}^j \mathbf{B}^k]] = \mathbb{E}[\text{tr}[\mathbf{A}^j]]\mathbb{E}[\text{tr}[\mathbf{B}^k]] \tag{13}$$

for any $j$, $k$, where tr is the trace.

Given a random unit vector $\mathbf{v}$, the expected squared lengths of the matrix vector products with $\mathbf{A}$ and $\mathbf{B}$ are given by

$$\mathbb{E}[\|\mathbf{A}\mathbf{v}\|^2] = \frac{1}{N}\mathbb{E}[\text{tr}[\mathbf{A}^2]], \ \mathbb{E}[\|\mathbf{B}\mathbf{v}\|^2] = \frac{1}{N}\mathbb{E}[\text{tr}[\mathbf{B}^2]] \tag{14}$$

The ratio $r_1$ of magnitudes is given by

$$r_1 \equiv \frac{\mathbb{E}[\|\mathbf{B}\mathbf{v}\|^2]}{\mathbb{E}[\|\mathbf{A}\mathbf{v}\|^2]} = \frac{\mathbb{E}[\text{tr}[\mathbf{B}^2]]}{\mathbb{E}[\text{tr}[\mathbf{A}^2]]} \tag{15}$$

Now consider the norm of $\mathbf{w} = \mathbf{A}^{-1}\mathbf{v}$ passed through each matrix:

$$\mathbb{E}[\|\mathbf{A}\mathbf{w}\|^2] = \frac{1}{N}, \ \mathbb{E}[\|\mathbf{B}\mathbf{w}\|^2] = \frac{1}{N}\mathbb{E}[\text{tr}[\mathbf{A}^{-1}\mathbf{B}^2\mathbf{A}^{-1}]] \tag{16}$$

The new ratio of magnitudes $r_2$ is given by

$$r_2 \equiv \frac{\mathbb{E}[\|\mathbf{B}\mathbf{w}\|^2]}{\mathbb{E}[\|\mathbf{A}\mathbf{w}\|^2]} = \mathbb{E}[\text{tr}[\mathbf{A}^{-1}\mathbf{B}^2\mathbf{A}^{-1}]] = \frac{\mathbb{E}[\text{tr}[\mathbf{B}^2]]}{\mathbb{E}[\text{tr}[\mathbf{A}^{-2}]]^{-1}} \tag{17}$$

From Jensen's inequality, $\text{tr}[\mathbf{A}^2] \geq \text{tr}[\mathbf{A}^{-2}]^{-1}$. Therefore $r_2 > r_1$; preconditioning by the inverse of $\mathbf{A}$ causes matrix-vector products to upweigh products with $\mathbf{B}$ relative to products with $\mathbf{A}$, relative to the unpreconditioned product.

In neural network settings, there are non-trivial relationships between the gradient, the Gauss-Newton matrix $\mathbf{H}_{\text{G}}$ and the functional Hessian $\mathbf{H}_{\text{F}}$; however, the eigenvectors and eigenvalues of $\mathbf{H}_{\text{G}}$ and $\mathbf{H}_{\text{F}}$ are only weakly related. Therefore even though the exact calculations in the example above don't hold, we suspect that generically preconditioning by $\mathbf{H}_{\text{G}}^{-1}$ will downweigh $\mathbf{H}_{\text{G}}\boldsymbol{\epsilon}^*$ compared to $\mathbf{H}_{\text{F}}\boldsymbol{\epsilon}^*$.

## C. Code Snippets for SAM and FUNCTIONAL-SAM

```python
from jax import grad, vjp
from jax.tree_util import tree_map
from utils import normalize_grad

def sam_gradients(params, loss_fn, rho):

    # compute the usual loss gradient
    dL_dtheta = grad(loss_fn)(params)

    # normalize the gradients
    dL_dtheta = normalize_grad(dL_dtheta)

    # perturb the parameters
    perturbed_params = tree_map(lambda a, b: a + rho * b, params, dL_dtheta)

    # compute the gradient as by SAM
    sam_grad = grad(loss_fn)(perturbed_params)

    return sam_grad

def functional_sam_gradients(params, loss_fn, network_fn, rho):

    # compute the usual loss gradient, but also extract dL_dlogits
    (dL_dlogits), dL_dtheta = grad(loss_fn, hax_aux=True)(params)

    # normalize the gradients
    dL_dtheta = normalize_grad(dL_dtheta)

    # perturb the parameters
    perturbed_params = tree_map(lambda a, b: a + rho * b, params, dL_dtheta)

    # set up the VJP at the perturbed parameters
    _, dF_dtheta_fn = vjp(lambda theta: network_fn(theta), perturbed_params)

    # do the VJP with the (unperturbed) dL_dlogits
    functional_sam_grad = dF_dtheta_fn(dL_dlogits)[0]

    return functional_sam_grad
```

Listing 1: Illustration of how to get the gradients in the two methods. Functional SAM differs from the SAM implementation only in the last couple lines

## D. ANGLE-SAM

In this section, we present a general variant of SAM, which includes both FUNCTIONAL-SAM and SAM as its particular instantiations. The core idea is that once we have been able to decompose the sharpness gradient into those arising from logit and functional paths, we can design our bespoke or custom versions of SAM which lean more or less towards one path than another.

To do so, let's assume that out custom path is at an angle $\phi$ with the functional path. We weigh the functional path by a factor of $\cos(\phi)$, while we weigh the logit path by $\sin(\phi)$. Then the gradient update in ANGLE-SAM can be described as:

$$\mathbf{g}_{\text{ANGLE-SAM}} := \nabla_{\boldsymbol{\theta}}\mathcal{L}(\boldsymbol{\theta}) + \sin(\phi) \cdot \mathbf{g}_{\mathtt{logit}} + \cos(\phi) \cdot \mathbf{g}_{\mathtt{func}} \tag{18}$$

$$= \nabla_{\boldsymbol{\theta}}\mathcal{L}(\boldsymbol{\theta}) + \sin(\phi) \cdot [\nabla_{\boldsymbol{\theta}}\boldsymbol{F}(\boldsymbol{\theta}) \cdot \nabla_{\boldsymbol{F}}\mathcal{L}(\boldsymbol{\theta} + \rho\boldsymbol{\epsilon}^*) - \nabla_{\boldsymbol{\theta}}\mathcal{L}(\boldsymbol{\theta})] + \cos(\phi) \cdot [\nabla_{\boldsymbol{\theta}}\boldsymbol{F}(\boldsymbol{\theta} + \rho\boldsymbol{\epsilon}^*) \cdot \nabla_{\boldsymbol{F}}\mathcal{L}(\boldsymbol{\theta}) - \nabla_{\boldsymbol{\theta}}\mathcal{L}(\boldsymbol{\theta})] \tag{19}$$

$$= \nabla_{\boldsymbol{\theta}}\mathcal{L}(\boldsymbol{\theta}) + \rho\sin(\phi) \cdot \mathbf{H}_{\text{G}} \cdot \boldsymbol{\epsilon}^* + \rho\cos(\phi) \cdot \mathbf{H}_{\text{F}} \cdot \boldsymbol{\epsilon}^* + \mathcal{O}(\rho^2) \tag{20}$$

We see that $\phi = \frac{\pi}{4}$ recovers SAM upto first order in $\rho$, while $\phi = 0$ would yield FUNCTIONAL-SAM and $\phi = \frac{\pi}{2}$ would result in a variant that optimizes solely along the logit path and which can thus be called LOGIT-SAM. *All in all, this shows how* ANGLE-SAM *is a clean generalization of* SAM*, that equips it with a perturbation angle in addition to the usual perturbation radius $\rho$.*

At some level, this above formulation could be viewed as making an assumption that these two paths are orthogonal[3]. On another level, one can simply think of these weights as a mere strategy to obtain convenient weight settings that have their sum of squares as unity.

*We expect that this approach might pay dividends in different model-dataset-optimizer triples, and we expect this to be an interesting direction for future work.*

---

[3]This is not far-fetched. As we noted in our experiments, these two paths tend to be anti-correlated, but often the correlation is quite small in magnitude and and approaches zero.

