# OpenReview forum: "Avoiding spurious sharpness minimization broadens applicability of SAM"
_ICML.cc/2025/Conference — ICML 2025 poster_

### Official Review · Reviewer_8e58 · 2025-03-03

**Overall Recommendation:** 3

**Summary:**

The authors investigate the Sharpness-Aware Minimization (SAM) algorithm for language tasks and find deteriorated performance compared to vision tasks. They explain this by re-writing the SAM update as a gradient norm penalty, and decompose the gradient of the gradient norm into a functional part and a logit part. Through empirical analysis they demonstrate that in language tasks, SAM is biased towards minimizing the logit part - different to vision tasks. They suggest a modified variant of SAM, which explicitly minimizes the functional part, and additionally contains a preconditioning for the perturbation. The authors report improved performance over baselines on language tasks with model sizes over three orders of magnitude (24M-1.2B).

**Claims And Evidence:**

- according to the title, the authors claim to “broaden the applicability of SAM”, implying that SAM becomes practical for (potentially large) language model optimization. However, the results presented in the experiments were conducted for a fixed number of steps, giving SAM twice the compute budget of base optimizers. The authors acknowledge this in their discussion, and point to efficient SAM implementations that could be combined with their work, but this is not explicitly shown. Thus, whether SAM's applicability is indeed broadened is unclear from this work.
- “Here, we confirm that the generalization benefits imparted by PRECOND FUNCTIONAL-SAM are brought about convergence to a solution with lower curvature, as shown in the Table 4 for the 23.9M model” This only holds when comparing SAM-variants to AdamW. In Table 6, the lowest curvature does not imply lowest eval loss (but still lower curvature compared to AdamW).
-  “This further highlights how SAM, by default, in language modeling tasks is set up to minimize sharpness spuriously”. The notion of spurious is a bit unclear here, as there is no way of assessing the non-spurious sharpness via the provided numbers, and PRECOND FUNC-SAM shows the lowest value for $tr(H_G)$. It could also just imply that overall the sharpness quantities investigated do not correlate well with generalization.

**Essential References Not Discussed:**

The authors provide a comprehensive discussion of the relevant literature.

**Experimental Designs Or Analyses:**

- like explained above, the difference in compute is problematic
- Since the authors aim at minimizing the functional path, it would be good to demonstrate that this actually happens with PRECOND FUNC-SAM, e.g. via repeating Figure 4 for _FUNC SAM (precond)_, and potentially also for the other variants
- Including _Func SAM_ and _SAM (precond)_ in Figure 3 would allow to disentangle the effects of preconditioning and  the functional formulation
- Since rho is tuned, I suggest reporting the full results in the Appendix (ideally like in Figure 3) for all experiments
- I recommend also reporting $\lambda_{max}(H_G)$ in Table 4

**Methods And Evaluation Criteria:**

Yes, except for the difference in compute budget between SAM and baselines.

**Other Comments Or Suggestions:**

- grammar mistake in line 186: “we see that a simple but spurious way to decrease it is by make the network…”
- typo in line 376: “interestingly, we also that”

**Other Strengths And Weaknesses:**

- the presentation of the paper is nice. I appreciate the effort in communicating the central ideas and results clearly (both through good writing and appropriate use of colors and markers)

- Given that there is a plethora of SAM papers and SAM variants, where each paper claims improved performance over baselines (standard SAM and standard optimizers), it is natural to ask why SAM is - to the best of my knowledge - not used for training LLMs. Investigating and improving the practicality of SAM for language tasks is therefore an important task, and the authors have made a good effort in this direction by finding a difference in sharpness-minimization between vision and language tasks, and proposing a modified SAM variant to mitigate the problems.

- The scale of the experiments (>1B training from scratch) is novel for SAM

- As discussed by the authors, other studies have also proposed preconditioning of SAM or adjusting its perturbation model, and the authors “believe our decomposition approach is orthogonal to this line of work”. While I agree that the perspective of decomposing the sharpness term is novel for SAM, it would still be interesting to see if some of the other SAM variants show the same behaviour as SAM or PRECOND FUNCTIONAL-SAM in language tasks. Perhaps some of the problems are already implicitly mitigated by those variants.

**Questions For Authors:**

I'll repeat points from above, in decreasing relevance
1. (most relevant) Do the authors have evidence that their method improves over baselines for the same compute budget?
2. Have the authors compared to other SAM variants? Could be for small-scale setups
3. Does the decomposition of the Hessian imply a non-spurious sharpness measure that might correlate better with generalization (e.g. via $\delta_{func}$)? If so, could the authors report it alongside the metrics in Table 4?
4. (least relevant) Do the authors have any intuition or evidence about where the difference between the language and vision setup and the corresponding spurious sharpness minimization comes from? I.e., why would one setup be biased towards the respective path? Can it be connected to the loss function, or the number of classes, or not training until convergence? Is there a connection to label smoothing? I'm just curious, and I don’t expect a comprehensive answer here; it will likely not affect my score, so the authors can feel free to not answer this.

**Relation To Broader Scientific Literature:**

SAM has mostly been applied to vision tasks, and to a lesser extent for fine-tuning in the language domain. Why it is not used more for language modelling in practice has not been investigated thoroughly. The authors show that training from scratch leads to deteriorated results compared to standard optimizers. This has not been demonstrated in published research. They connect their findings to previous work that highlighted the relevance of the Nonlinear Modeling Error (NME), a component of the Hessian of the loss, for Sharpness minimization. They derive a SAM-variant that explicitly minimizes the functional part, and additionally employs a preconditioning on the SAM perturbation. The preconditioning of the SAM perturbation is conceptually similar to the plethora of SAM variants that exists already. Those variants are discussed, but not compared against in the experiments. The paper is also the first one I am aware of that scales SAM to models bigger than 1B params.

**Theoretical Claims:**

I checked Appendix B.

---

> ### Author Rebuttal · Authors · 2025-04-01
>
> We thank the reviewer for their positive assessment, insightful comments, and detailed feedback. We are glad that you were able to contextualize our contribution quite spot on.
>
> &nbsp;
>
> ----
>
> &nbsp;
>
>
> > ###  Same compute budget comparisons
>
> As it stands, at an equal number of FLOPs, well-tuned AdamW does slightly outperform  functional SAM. However, there are nevertheless important reasons why we believe functional SAM is still promising:
> \
> &nbsp;
> - **Relevant Use Cases:** There are practically relevant use cases, such as in data-limited regimes or where the model size is constrained (e.g. due to inference time constraints), where the better performance at fixed step count is desirable and where the extra training overhead of functional SAM may not matter as much.
> \
> &nbsp;
> -  **Solution Quality Beyond Loss:** Even in the equal-FLOP setup, functional SAM’s better geometric properties in its flatter solution might be preferable than a method yielding sharper solutions, since it has been well-documented [Liu et. al., 2023] flatness of the solution correlates more robustly with downstream performance than similar values of loss.
> \
> &nbsp;
> - **Path to Efficiency:** Lastly, functional SAM is algorithmically compatible with efficient SAM approaches. Approaches like LookSAM [Liu et. al., 2022] can decrease the overhead of SAM to 5-10% while maintaining much of the benefit of the method. We hope to test this and other approaches in future work, and we believe that the overhead can be greatly reduced.
>
> &nbsp;
>
> > ### Comparison to other SAM variants:
>
> - While interesting, comparing against the plethora of (effectively vision-focused) SAM variants was beyond the scope of this work.
> \
> &nbsp;
> - During our initial investigation into SAM's poor performance in LMs, we did experiment with common, simpler variations, such as using the unnormalized perturbation step or trying different weighted combinations of the SAM gradient and the standard gradient. However, these modifications did not appear to fundamentally resolve the performance degradation observed in language modeling tasks.
> \
> &nbsp;
> - In retrospect, this is perhaps not entirely surprising.  Critically, *none of these existing variations are explicitly designed to prioritize functional sharpness minimization over logit sharpness*. Our diagnosis revealed that this preferential treatment of functional geometry is precisely what is needed to make SAM effective in LMs, requiring a more significant departure from standard SAM formulation.
> \
> &nbsp;
> - It would be interesting to nevertheless explore other orthogonal SAM variants, which address issues like parameter scaling sensitivity (ASAM, Kwon et al., 2021), norm choices (Tahmasebi et al.,2024) or perturbation stability (ESAM, Li & Giannakis, 2024), in relation to functional SAM as well. But since the design space of new algorithms tends to explode, we have had to stay within our scope making SAM effective in pre-training LMs.
>
>
> &nbsp;
>
> > ### Non-spurious sharpness measures that might correlate better with generalization
>
> This is an excellent remark. We do think that something which directly measures the *extent of functional curvature* (like the frobenius norm of the functional Hessian) *or its relative extents compared to the logit curvature* to potentially be revelatory. We will try to add some measurements of this kind in the final version, but this more likely deserves a separate study of its own.
>
> &nbsp;
>
> > ### Vision vs Language Intuition:
>
> Another great question. Our current hypothesis relates to the **nature of the typical output distributions $p(y | x; \theta)$ in these domains**.
>
> - In many **vision** tasks, the probability mass often *concentrates over a relatively small number of semantically related classes* or visual scenes. The output distribution might be less dispersed, and *manipulating logit statistics could potentially align reasonably well with improving the underlying function's robustness*.
> \
> &nbsp;
> - In **language modeling** (specifically next-token prediction), the **distribution over the next token is often highly dispersed and heavy-tailed**, with non-negligible probability assigned to many different words. In such a setting, minimizing sharpness simply by manipulating logit statistics (e.g., making the distribution slightly peakier) might be an *"easy" path for the optimizer that doesn't translate to genuine improvements in the functional geometry*.
>
> &nbsp;
>
> > ### Suggestions on Experimental Designs Or Analyses
>
> - Thanks for these great suggestions, we will definitely incorporate them in the camera ready version.
>
> &nbsp;
>
> ----
>
> *We hope we have able to address your concerns. We remain at your disposal should you have more questions or comments.*

---

> > ### Comment · Reviewer_8e58 · 2025-04-02
> >
> > I would like to thank the authors for responding to my comments, and for the insights about vision vs language and non-spurious sharpness measures. I still believe that the overall direction of this paper is nice, but my stance regarding the main points (compute budget, other SAM variants) does not change in light of the rebuttal. I will therefore keep my score.

---

### Official Review · Reviewer_Eee3 · 2025-03-11

**Overall Recommendation:** 3

**Summary:**

This paper presents an intriguing exploration of the distinction between logit-space and functional-space perturbations within the context of Sharpness-Aware Minimization (SAM). The authors' identification of this subtle difference is interestimng, and while the observed effects might appear minor, the potential ramifications for model training and generalization are substantial. Understanding how these perturbation spaces impact optimization could lead to more robust and efficient training methodologies.

## update after rebuttal
thanks for addressing some of the concerns. I think this work has merit and I will keep to my original score [weak accept]

**Claims And Evidence:**

The authors propose P-SAM to address issues related to "un-preconditioned geometry." However, it appears that the concept of pre-conditioning the inner adversarial optimizer with the outer optimizer's state, specifically using ADAM, is already a common practice, notably within the Jax's Optax library - and in other works (like Granziol, JMLR; Gordon-Wilson and others).

If the authors' P-SAM simply replicates this existing approach, then the novelty and contribution are limited. If, on the other hand, P-SAM introduces "further preconditioning" beyond established techniques, the justifications provided in the paper are insufficient. The explanations regarding the need for additional pre-conditioning lack the necessary depth and clarity to convince of necessity or effectiveness. More concrete theoretical or empirical evidence is needed to substantiate this claim and differentiate P-SAM from existing implementations.

While the authors suggest that F-SAM should deliver superior performance, the empirical evidence provided is underwhelming. A mere 0.03 improvement in loss, based on what appears to be a single seed and a fixed training budget, raises serious questions about statistical significance. Without a clear understanding of the variance across multiple seeds under identical training conditions, it's difficult to ascertain whether this improvement is genuinely meaningful or simply a by-product of experimental fluctuation. For typical NLP problems, where variability between runs can often be considerable, a 0.03 difference might well fall within the noise - this should be commented on. To validate the effectiveness of F-SAM, surely a more rigorous experimental setup, including multiple seeds and a thorough analysis of variance, is essential? Furthermore, it would be beneficial to benchmark these improvements against typical NLP problem improvements to provide more context.

Adding to my concern is the absence of F-SAM results in Figure 2. This would have been more informative had they included F-SAM.

**Essential References Not Discussed:**

The key innovation of the paper is the exploration of the distinction between logit-space and functional-space perturbations within the context of Sharpness-Aware Minimization. I could not find prior work that replicates this

**Experimental Designs Or Analyses:**

coming back to the issue of the headline result, which seems to be a small improvement with lack of clarity about its significance. as per previous comments, some multi-run analysis would be good to see if this [slight] improvement is truly significant.

**Methods And Evaluation Criteria:**

The general methods suggested make sense for the application - but, as per my comments, there are significance questions about the improvements.

**Other Comments Or Suggestions:**

The paper is well-written and an enjoyable read with no obvious errors or typos. The references need Capital letters protecting [minor issue].

**Other Strengths And Weaknesses:**

The core novelty of the paper is nice -  logit-space and functional-space SAM. This could have some neat practical implications. While the observation of the logit/functional difference is an interesting contribution, the paper's central claims regarding F-SAM and P-SAM are weakened by apparent methodological shortcomings and a lack of empirical support. To strengthen the paper, the authors should address the (seemingly weak) statistical significance of their results, provide a more comprehensive experimental evaluation and offer a clearer justification for the proposed pre-conditioning method and how it differs from existing approaches.

**Questions For Authors:**

all in previous comments

**Relation To Broader Scientific Literature:**

there is a other of work in pre-conditioning, some of which is referenced in the paper - but I'd argue that's not the headline of the submission [as pre-conditioning by itself is not novel]. There are some prior works that might be of interest.
A Random Matrix Theory Approach to Damping in Deep Learning, Diego Granziol, Nicholas Baskerville [arXiv]. Granziol's JMLR paper also looks at a Hessian pre-conditioning [using RMT] and there is related work from Gordon-Wilson and Izmailov and Das's arXiv paper Towards Quantifying the Preconditioning Effect of Adam.

**Theoretical Claims:**

all proofs checked (to the best of my ability) and all seem to work out.

---

> ### Author Rebuttal · Authors · 2025-04-01
>
> We thank you for your feedback and sharing the interesting works. Besides, we are pleased to hear that you find the exploration intriguing and recognize its potentially substantial ramifications.
>
> &nbsp;
> ----
>
> > ### 1. Significance of Empirical Gains (0.03 loss):
>
> We understand the concern about seemingly small improvements in loss. However,
>
> - **Context is Key**: In large-scale LM pre-training (100M-1B+ params), **improvements of 0.03-0.06 validation loss** (Tables 1, 2, 3) are **practically meaningful**. They are *comparable to or exceed gains in well-respected works on LLM optimizers* (e.g., SOAP [Vyas et al., 2024], CASPR [Duvvuri et al., 2024] which are both improvements over Shampoo [Gupta et al., 2018] and report similar gains if not lower) and newer variants of Attention [Leviathan, et. al., 2025], LLM Fusion [Mavromatis, et. al. 2024], and corpus deduplication [Fig Lee, et. al., 2022],  to list a few. All of these mentioned papers consider the C4 dataset, and so the differences are comparable.
> \
> &nbsp;
> - **Statistical Significance:**
>    - As discussed in Lines 263-274 of Section 5.2, for prototyping we conducted our experiments using **3 random seeds**. We observed that validation loss results were typically stable to the **3rd or 4th decimal place**, indicating very low variance between runs. The reported results in Table 3 are averaged over 3 seeds (where the various methods rank as, 3.86 vs 3.88 vs 3.90 in Table 3 for precond. Functional SAM vs Functional SAM vs AdamW) are therefore highly significant and not due to noise. We will clarify this.
> \
> &nbsp;
>    - For the later experiments at the much larger parameter scale, we did have to report single seeds due to the constraints of time and corresponding costs of these experiments. However, to address this valid concern of yours, we have carried out an experiment on the **1.2 B parameter model over 3 seeds**, and the averaged results in a fixed-length (50K step) settings are **precond. Functional SAM 2.70** versus **AdamW 2.73**. This highlights that our results continue to be statistically significant even at these scales. We will include these results in the revision.
> \
> &nbsp;
>
> Hence, we can be confident that the gains delivered through our method are genuinely meaningful. In fact, it would serve us to remember that standard SAM *consistently performed worse* than AdamW (Fig 1). We have been able to turn things around and shown, for the first time, *positive* gains from SAM-style regularization in this setting over AdamW. And, *the difficulty of achieving any improvement over tuned AdamW at this model scale cannot be overstated.*
>
> &nbsp;
>
> > ### 2. Preconditioning Novelty (vs. Optax, Granziol, etc.):
>
> We appreciate the reviewer pointing out related work and common practices.
>
> - Our contribution regarding preconditioning (Sec 4.2) should be understood specifically as addressing the potential **mismatch between SAM's default Euclidean perturbation** and the preconditioned geometry used by the *outer optimizer* (AdamW), particularly relevant for heterogeneous Transformer landscapes. Moreover, we also provide a theoretical argument (App B.1) that preconditioning can help **re-balance logit/functional paths**, which enriches our perspective about preconditioning as well.
> \
> &nbsp;
> - While general preconditioning and using Adam's state (as perhaps done implicitly in some Optax implementations) are known, what constitutes novelty in this specific context is the **explicit motivation for fixing SAM's failure in LMs** by aligning geometries/rebalancing paths, and the *demonstration of its effectiveness especially in combination with functional SAM* (Table 3 shows precond. functional SAM outperforms plain functional SAM and precond. SAM).
> \
> &nbsp;
> - Works like Granziol et al., Das et al., while quite intriguing, explore preconditioning for the main step, **not specifically for SAM's perturbation**. We will refine Sec 4.2 and Related Work to better delineate our specific contribution versus existing preconditioning concepts.
>
> &nbsp;
>
> ----
>
> Hopefully, this addresses your pending concerns, but please let us know if you have any more questions or comments.

---

> > ### Comment · Reviewer_Eee3 · 2025-04-06
> >
> > Thank you for addressing my concerns. Clarification of the significance of the results would help a prospective reader, as would making clear how the approach you suggest differs from other works readers may be familiar with. I will keep my score - I do think this is potentially interesting work, with several possible extensions.

---

### Official Review · Reviewer_ewUR · 2025-03-12

**Overall Recommendation:** 2

**Summary:**

The paper investigates the limitations of SAM in NLP tasks, where it often degrades performance despite its success in vision tasks. The authors find that SAM's effectiveness varies across domains due to differences in sharpness minimization pathways: the logit path and the functional path. In NLP, the logit path dominates, leading to spurious sharpness minimization.
The paper proposes two alternative algorithms: Functional SAM and Preconditioned SAM. Empirical evaluations demonstrate improved performance over AdamW and SAM in NLP tasks.

**Claims And Evidence:**

See Weakness.

**Essential References Not Discussed:**

Some improved algorithms of SAM:

[1] Du et al. Efficient sharpness-aware minimization for improved training of neural networks. (ICLR 2022)

[2] Mueller et al. Normalization layers are all that sharpness-aware minimization needs. (NeurIPS 2023)

[3] Wang et al. Improving generalization and convergence by enhancing implicit regularization. (NeurIPS 2024)

**Experimental Designs Or Analyses:**

See Weakness.

**Methods And Evaluation Criteria:**

See Weakness.

**Other Comments Or Suggestions:**

See Weaknesses.

Typo: (Line 377) "We also that"

**Other Strengths And Weaknesses:**

**Strengths**
- The paper presents a novel decomposition of SAM’s sharpness minimization update into logit and functional paths, revealing that the logit path dominates in NLP tasks.
- The proposed algorithms, Functional SAM and Preconditioned SAM, empirically outperform SAM in certain NLP tasks.


**Weaknesses**
- **Computational Overhead**: Similar to SAM, Functional SAM and Preconditioned SAM require **twice** the gradient computation per step, making them computationally expensive. Consequently, while the proposed algorithms slightly outperforms Adam *given the same number of iterations*, Adam may still perform better when compared *under equal computational cost*, which is a fairer comparison.

- **Concern regarding long-term performance.** *It is unclear whether the proposed algorithms will be surpassed by AdamW or SAM given sufficient training time.* The experiments on C4 are conducted for a relatively small number of steps. Even with a 1.2B model, the final validation loss remains above 3, which is **significantly higher** than established baselines. For instance, in [4], a 1.2B model achieves a final validation loss of 2.56.

- **Insufficient experimental evidence** to explain how the proposed algorithms work. Although the proposed algorithms are designed to enhance the functional path, no experiments demonstrate whether they indeed result in a larger functional path than SAM.

- **Insufficient theoretical support.**  While the proposed algorithms are motivated by addressing the domination of the functional path,their formulation relies on several approximations. Theoretical support is needed to substantiate that they indeed lead to a larger functional path, particularly for Preconditioned SAM (, whose connection to the main motivation remains unclear).



[4] Zhao et al. Deconstructing What Makes a Good Optimizer for Autoregressive Language Models. (ICLR 2025)

**Questions For Authors:**

See Weaknesses.

**Relation To Broader Scientific Literature:**

See Summary.

**Theoretical Claims:**

N/A

---

> ### Author Rebuttal · Authors · 2025-04-01
>
> We thank the reviewer for their detailed comments and address some of their concerns here.
>
> &nbsp;
>
> ----
>
> &nbsp;
>
> > ### 1. Computational Overhead:
>
> We agree that in its current form, Functional SAM is not as FLOPs efficient as Adam.
>
> - However, FLOPs are not the only limiting factor in training; for example, in certain scenarios industrial practitioners tend to be **data limited** or **model size limited**. In these scenarios, *extra training time is acceptable for better final quality*, and functional SAM can be a better fit.
> \
> &nbsp;
> - In addition, functional SAM is compatible with efficiency techniques like LookSAM [1]; this literature suggests that we may be able to reduce the overhead to 5-10% while maintaining most of the benefit of (Functional) SAM. We hope to pursue this avenue in future work.
> \
> &nbsp;
> - Our focus here was establishing the *effectiveness* of the functional path approach first. Our work provides a crucial understanding of SAM's limitations and offers the first validated approach to successfully apply sharpness-aware methods to large-scale LM pre-training (as noted by **Reviewer 8e58**).
>
> [1] https://arxiv.org/abs/2203.02714
>
> &nbsp;
>
> > ### 2. Concern regarding long-term performance:
>
> Please have a look at *Table 3*, where we already show the gains provided by functional SAM sustain longer training durations as well. **Our 1.2 B model gets 2.61 in terms of validation loss**, which is very close to the 2.56 validation loss from [4] which you have alluded to. These 0.05 differences between the two works can easily be because their empirical setups, such as hyperparameters, exact architectural implementation, might not be identical.
>
> *Thus, we can be confident that functional SAM does yield long-term performance benefits as well.*
>
> &nbsp;
>
> > ### 3. Demonstrating Enhanced Functional Path:
>
> This is a great suggestion, and we are working on measurements to show this. The experiments did not finish in time for the rebuttal but will be included in the revision.
>
> &nbsp;
>
> > ### 4. Theoretical Support and Preconditioned SAM Motivation:
>
>   - **Functional path formulation:** The functional SAM update (Eq. 11) itself **involves no approximation, and is an exact analogue of the original SAM update (Eq. 2)**, but where the contributions along the logit-sharpness path have been suppressed by design. We used the penalty SAM formulation for discussion following previous works which take advantage of the fact that penalty SAM is more amenable to theoretical analysis while simultaneously giving similar performance to original SAM. This gave us easier way to present and delineate the differences between SAM and Functional SAM.
> \
> &nbsp;
>  - **Preconditioned SAM:** The motivation is twofold:
>     - (1) Empirically motivated: To address the mismatch between SAM's spherical perturbation and AdamW's elliptical perturbation due to its diagonal preconditioning, which might cause issues in heterogeneous landscapes like Transformers.
>     \
> &nbsp;
>      - (2) Theoretically motivated: Preconditioning the perturbation by approx. $H_{G}^{-1}$ (approximated by AdamW's ${M}^{-1}$) can selectively dampen the logit path ${\delta}_{logit} = H_G \epsilon^\ast$ more than the functional path $\delta_{func} = H_F \epsilon^*$, thus promoting the functional path (detailed in App B.1, where we made basic assumptions to make the argument quantitative). We will clarify this motivation in Sec 4.2.
>
> &nbsp;
>
> > ### 5. Missing References [1-3]:
>
> - [1] and [2] are related to SAM in that they propose more efficient variants [1] or modifications based on normalization layers [2]. But both of these are exclusively evaluated in vision, where they do not interface with the problem faced by SAM in language modeling tasks.
> \
> &nbsp;
> - Although ref [3] shares similar motivations as SAM and is quite interesting, they in their own words say that the “specific approaches differ significantly”.
> \
> &nbsp;
> - These works are firmly orthogonal to our core contribution: diagnosing the *specific failure mode* of SAM in LMs (logit path dominance) via a novel decomposition and proposing targeted fixes (Functional SAM, precond. SAM) that make SAM *effective* in this domain for the first time.
>
> Regardless these will be good works to discuss in our paper, and we thank you for suggesting them.
>
> ----
>
> &nbsp;
>
> *Let us know if we can clarify any further points. If we have answered your concerns, **please consider giving your score a second thought**.*

---

### Official Review · Reviewer_TL8s · 2025-03-18

**Overall Recommendation:** 1

**Summary:**

The paper introduces Functionnal-SAM (F-SAM), an alternative to Sharpness-Aware Minimization (SAM) that aims to address its poor performance in NLP tasks. The authors argue that SAM's failure in language modeling is due to its focus on regularizing logit statistics rather than modifying the functional properties of the neural network. They propose F-SAM, which modifies sharpness through the functional path, and PRECONDITIONED-SAM (Pre-SAM), which improves SAM’s perturbation by adapting it to the optimizer’s preconditioning scheme. Their empirical results show very slightly improved performance over both SAM and ADAMW across multiple model scales and training settings.

**Claims And Evidence:**

The main claims of the paper are:
- SAM performs poorly in NLP tasks because it minimizes sharpness primarily by modifying logits statistics rather than the network’s functional properties.
- F-SAM improves sharpness regularization by emphasizing functional modifications over logit-based adjustments.
- Pre-SAM further improves sharpness minimization by adapting perturbations to the optimizer’s preconditioning scheme.

The combination of F-SAM and Pre-SAM shows slight performance improvement (max improvement displayed is 0.06 loss points on values around 3.5) in large-scale language modeling.

The evidence includes:
- Theoretical decomposition of sharpness minimization into logit and functional paths.
- Empirical validation of the proposed algorithms on multiple model scales (from 2M to 1.2B parameters) in both fixed-length and Chinchilla-style training regimes.
- Hessian eigenvalue analysis showing F-SAM reduces sharpness more effectively than SAM.

However, key claims remain heuristic rather than rigorously proven, and the cost-benefit trade-off is not discussed in sufficient depth.
Further, the empirical results show only *very* marginal improvements over SAM and more critically ADAMW, raising questions about the practical significance of the proposed method.

**Essential References Not Discussed:**

N/A

**Experimental Designs Or Analyses:**

The experimental setup is very clear and sufficient in terms of dataset and model choices, but it could be improved in several ways:
- Computational cost is not analyzed, making it unclear whether F-SAM is worth the additional cost.
- No training time comparisons between F-SAM, SAM, and ADAMW.
- Limited discussion on efficiency: if F-SAM is significantly more expensive while offering small improvements, it is not practically useful.
- The results are underwhelming in terms of performance improvements, with the best improvement being 0.06 loss points on values around 3.5. This raises questions about the practical significance of the proposed method.

**Methods And Evaluation Criteria:**

The proposed methods are evaluated using:
- Validation loss on language modeling tasks with multiple model scales.
- Hessian eigenvalue analysis to assess sharpness reduction.
- Performance comparisons with SAM and ADAMW under equivalent computational budgets.

The evaluation is generally well-structured but has critical weaknesses:
- The efficiency trade-offs of F-SAM and Pre-SAM are not analyzed and are merely discussed in Section 7.
- No training time comparisons or FLOP analysis to assess whether F-SAM justifies its additional computational cost.
- No comparisons to alternative sharpness minimization methods beyond SAM, leaving open the question of whether F-SAM is the best solution for this problem. Further, the claim about improved versions of SAM (Kwon et al., 2021; Tahmasebi et al.,2024; Li & Giannakis, 2024) being an orthogonal line of work is not substantiated and would benefit from either a more detailed discussion or empirical comparison.

**Other Comments Or Suggestions:**

N/A

**Other Strengths And Weaknesses:**

**Strengths:**
- Identifies a limitation of SAM in NLP and proposes an alternative.
- Provides a novel decomposition of SAM into logit and functional paths.
- Coherent empirical framework across a wide range of model scales.
- Hessian eigenvalue analysis to support empirical claims is given

**Weaknesses:**
- No formal proof of F-SAM’s theoretical advantages.
- The computational efficiency is not analyzed, making practical applicability uncertain.
- The empirical results are very weak and would not justify the additional cost of F-SAM.

**Questions For Authors:**

- Are the proposed methods adapted for fine-tuning settings? Furthermore, do the given pre-trained models lead to better zero/few-shot performance on downstream tasks?
- Would the proposed methods be compatible with parameter-efficient fine-tuning techniques such as LoRA or Adapters? Could they be combined in a fine-tuning pipeline?
- SAM is often seen as a regularization technique, however, it is not explained in the paper how F-SAM and Pre-SAM relate to explicit regularization techniques. Could they be combined with L1/L2 regularization or other explicit regularization methods to improve performance further?
- The Hessian metrics reported are not associated with their variance but are known to be empirically noisy. Could you provide more details on the Hessian analysis and how it was conducted?

If the overall weaknesses and questions are addressed, I would be happy to raise my score, although I believe the paper would benefit from more substantial improvements (Angle-SAM seems to be a very promising theoretical approach) to justify a higher rating.

**Relation To Broader Scientific Literature:**

The paper is very well-situated within the sharpness regularization literature, and even in the preconditionned optimization literature, even though one could regret the absence of preconditioning-based optimization methods like Shampoo (a precursor of SOAP by Vyas et al.) or [K-FAC](https://arxiv.org/abs/1503.05671).

**Theoretical Claims:**

The paper presents a decomposition of sharpness minimization into logit and functional paths but does not provide a rigorous proof that F-SAM leads to better generalization. Instead, the claims are supported by empirical observations and qualitative reasoning which are unfortunately not sufficiently backed up by experimental results in my opinion.

Unfortunately, there is no formal proof that logit-path minimization is suboptimal for NLP. Furthemore, the paper does not provide a theoretical justification for why F-SAM gives better convergence guarantees than SAM.

The appendix introduces ANGLE-SAM, which generalizes SAM by parameterizing perturbations using an angle $\phi$, showing that F-SAM and SAM are special cases. However, this remains an intuitive generalization rather than a rigorous theoretical result. I believe exploring such a theoretical generalization and studying its properties with respect to $\phi$ could be a strong contribution to the paper.

---

> ### Author Rebuttal · Authors · 2025-04-01
>
> We thank you for your thorough review. We address the primary concerns below:
>
> &nbsp;
>
> ----
>
> &nbsp;
>
> > ### 1. Significance of Performance Improvements:
>
> - In LLM pre-training (100M-1B+ params), achieving consistent validation loss improvements of 0.03-0.06 (as seen in Tables 1, 2, 3) is **highly significant**. This magnitude is comparable to or *exceeds* gains reported in recent, well-regarded optimizers for LLMs, such as SOAP [Vyas et al., 2024, 0.04-0.06 improvement] and CASPR [Duvvuri et al., 2024, ~ 0.01-0.03]. This is also similar to the size of gains from architectural and data processing changes [Leviathan, et. al., 2025, Mavromatis, et. al. 2024, Fig Lee, et. al., 2022].
>
> - Crucially, **prior to our work, SAM consistently *degraded* performance** compared to AdamW in this setting (Fig 1). Our methods are the first to successfully leverage SAM-style regularization for *improved* LM pre-training results across scales, representing a notable advance in the field.
>
> -  As clarified in our response to Reviewer Eee3, these gains are statistically significant and indeed represent genuinely meaningful improvements.
>
> &nbsp;
>
> > ### 2. Computational efficiency concerns
>
>  - *Costs relative to SAM and AdamW:* functional SAM has virtually identical computational cost (FLOPs and memory) as standard SAM. Both require one forward pass, one backward pass for the initial gradient, and one backward pass (VJP) for the SAM/F-SAM gradient, resulting in ~2x the cost of AdamW per step. We will include explicitly measured time per step in the revision.
>
> - *Fixed Data Budget & Model Size scenarios:* Our primary comparison point in the paper is *equal steps* since this is practically relevant in scenarios limited by data availability or required model size (e.g., inference constraints), where extra training time is acceptable for better final quality.
>
> - *Future Efficiency:* As discussed in Section 6, Functional SAM is compatible with efficient SAM methods like LookSAM [1], offering a clear path to reducing the overhead to ~5-10% in future work. Our focus here was the fundamental advance of making *any* SAM variant work effectively with language models, as also identified by Reviewer 8e58.
>
> &nbsp;
>
> > ### 3. Comparison to Other Methods:
>
> - *Improved SAM Variants:* Methods like ASAM (Kwon et al., 2021), ESAM, etc., primarily target *vision* tasks and do not address the fundamental logit-path dominance issue we identified in *language modeling*. Our decomposition and Functional SAM are thus orthogonal contributions aimed at *fixing SAM's failure in a new domain*.
>
> - *Preconditioning Methods (Shampoo/K-FAC):* Our preconditioned-SAM technique is indeed compatible with any preconditioning method, and it would be interesting to try our technique with base optimizers which use non-diagonal preconditioning. We leave this to future work.
>
> &nbsp;
>
> > ### 4. Theoretical Claims:
>
>  - **Proofs:** We followed a common paradigm in deep learning research: identify an empirical issue, propose a diagnostic (logit/functional path), develop a principled fix (functional SAM), and validate empirically. Rigorous proofs for SAM-like methods are challenging and sometimes unhelpful (see response to X2he, point 2). Our consistent empirical gains across scales strongly support our hypothesis.
>
> - **Angle-SAM:** We appreciate the reviewer's interest in Angle-SAM, which arises naturally from our decomposition. We de-emphasized it in the current work because it did not add additional benefits for our primary goal of making SAM effective for LLMs.
>
> &nbsp;
>
> > ### 5. Experimental Details (Hessian Variance):
>
> Hessian metrics (Table 4/App Table 6) were computed using standard techniques (e.g., Lanczos for $\lambda_{max}$, Hutchinson for trace) averaged over 50 batches from the validation set, where each batch is 256 sequences of length 512, and thus **these metrics are aggregated over ~6.5 million tokens**. We noticed that even as few as 5-10 batches already gave stable results, but in our Tables we report it with 50 batches for additional precision.
>
> &nbsp;
>
> > ### 6. Specific Questions:
>
> - *Fine-tuning/Zero-shot:* Future work, but flatter minima (which we achieve, Table 4) often correlate with better transfer and robustness [Liu et al., 2023]. Pruning results (Fig 5) also suggest improved robustness.
> - *PEFT Compatibility:*  Likely compatible, but the interaction needs study.
> - *Explicit Regularization (L1/L2):* Yes, func. SAM can be seen as a regularizer & is compatible with like L1/L2 regularization (we already use Weight Decay of 0.1)
>
> &nbsp;
>
>
> We believe our paper offers a novel diagnosis and the first effective solution for applying SAM to large-scale LM pre-training, a significant and previously unsolved problem. The empirical gains are meaningful in this context and demonstrate the success of our approach.
>
> ----
>
>
> *Let us know if you have additional questions; if we have answered your concerns, we hope you will consider revisiting your review score.*

---

### Official Review · Reviewer_X2he · 2025-03-19

**Overall Recommendation:** 3

**Summary:**

This paper investigates why Sharpness Aware Minimization (SAM), effective in vision tasks, underperforms in natural language processing (NLP). The authors identify that SAM in NLP overly focuses on reducing sharpness via logit manipulation rather than improving the model's functional geometry, leading to spurious optimization. To address this, they propose Functional-SAM, which prioritizes functional sharpness reduction, and preconditioned SAM, aligning perturbations with optimizer geometry, demonstrating superior generalization across NLP tasks and model scales compared to SAM and AdamW.

**Claims And Evidence:**

While the logit vs. functional sharpness decomposition is intuitive, the theoretical justification relies heavily on empirical observations and simplified assumptions (e.g., free independence of Hessian components). A more rigorous mathematical foundation for the decomposition’s validity across architectures and loss landscapes is lacking.

**Essential References Not Discussed:**

The paper references [1], which analyzes gradient norm penalties in the context of SAM sharpness. To clarify the novelty and distinctions of this work, could the authors explicitly discuss how the decomposition of the sharpness gradient in their approach differs from that in [1]?

[1]Penalizing Gradient Norm for Efficiently Improving Generalization in Deep Learning. ICML2022

**Experimental Designs Or Analyses:**

1.The evaluation primarily focuses on language modeling using the C4 dataset and decoder-only Transformers. The paper does not validate the proposed methods (Functional-SAM and preconditioned SAM) on other NLP tasks (e.g., text classification, machine translation) or diverse datasets, raising questions about broader applicability.

2.Reliance on C4 dataset alone limits insight into performance on noisy or domain-specific corpora.

**Methods And Evaluation Criteria:**

Downstream utility (e.g., fine-tuning, robustness) is only briefly explored (via pruning), leaving practical NLP benefits underdeveloped.

**Other Comments Or Suggestions:**

To ensure reproducibility and facilitate further research, could the authors provide access to the implementation code?

**Other Strengths And Weaknesses:**

The methods inherit SAM’s 2× computational overhead, and the perturbation radius requires careful tuning, especially for large models. The paper notes that tuning \rho becomes coarser for billion-parameter models, potentially limiting real-world adoption where hyperparameter optimization is costly. Additionally, combining Functional-SAM with preconditioning introduces more complexity, which may hinder ease of use.

**Questions For Authors:**

see questions before.

**Relation To Broader Scientific Literature:**

1.This paper provide empirical insights into sharpness regularization.


2. Based on their findings, authors propose Functional-SAM and Preconditioned SAM

**Theoretical Claims:**

lacks of theoretical guarantees.
1.No convergence analysis for Functional-SAM or preconditioned SAM.
2.No theoretical bounds on how much functional sharpness reduction improves generalization.

---

> ### Author Rebuttal · Authors · 2025-04-01
>
> We thank the reviewer for their detailed comments and their positive view of our paper. We address their concerns below
>
>
> ----
>
> &nbsp;
>
>
> > ### Theoretical justification of the decomposition
>
> - The decomposition into logit vs. functional sharpness is valid in *any setup involving the composition of a loss function with the outputs of a parameterized function*.
> \
> &nbsp;
> - This describes virtually all scenarios in deep learning, and there is **fundamentally nothing which limits its validity to a specific architecture**. We will clarify this aspect in the text to avoid any confusion.
>
> &nbsp;
>
> > ### Lack of theoretical guarantees
>
> - **Convergence Analysis:** We appreciate the comment. However, rigorous convergence analysis for SAM-type methods is notoriously complex:
>   - SAM itself has lacked a general convergence proof in the non-convex case, with initial results only recently published [8].
>   - Older convergence analyses [1-5] required significant modification of the algorithm itself or strong assumptions to make progress.
>    - The utility of convergence proofs for the design of SAM-like algorithms is also debatable. There is empirical and theoretical evidence that SAM is regularizing sharpness throughout training, rather than just selecting final flat minima to converge to in either the early or late time dynamics only [6].
> \
> &nbsp;
> -  **Generalization Bounds:** we note that **none of the existing generalization bounds can account for the ineffectiveness of SAM in language modeling**. This observation highlights the potential *pitfalls of pursuing generalization bounds* without adequately accounting for the impact of optimization dynamics.
>
> &nbsp;
>
> > ### Concerns about efficiency and deployment of methods
>
> **Computational Overhead:** We agree that the computational overhead needs to be improved; we believe that SAM efficiency methods like LookSAM [7] should be compatible with Functional SAM and can reduce the overhead to a more modest 5-10% — which we hope to demonstrate in future work.
>
> - **Rho $\rho$ Tuning:** for large models, $\rho$ can be added to scaling studies already used for other hyperparameters (learning rate, weight decay) using detailed experiments at small scales to predict good hyperparameter values at large scales.
>
> &nbsp;
>
> > ### Reliance on C4
>
> - C4 is widely used for benchmarking LLMs and using it facilitates easier comparison to those works.
>
> - Additionally, as mentioned in the paper, C4 is a clean dataset and thus a hard testground for regularization techniques, like SAM, that aim to improve generalization. *We expect the gains to be even higher when the dataset is noisy.*
>
> - Besides, C4 is a gigantic dataset significant coverage of most textual corpora on the internet. Thus, the evaluation here being dataset-specific is much less of a risk.
>
> &nbsp;
>
> > ### Decoder-only Transformers
>
> We focus on this setting to be closer to the industrial use-case, as **decoder-only Transformers are really the workhorse of generative LLMs**. Downstream tasks can all be modeled on top of these decoder-only models, say, via in-context learning.
>
> &nbsp;
>
> > ### Practical NLP benefits underdeveloped
>
> We understand your concern, and reiterate that before this work, there was not even a clear path to making SAM effective on language tasks. We believe our work has demonstrated a viable path, and hope to develop a truly practical version of the method in future work. We also recommend you to check Reviewer 8e58’s remarks, where they attest to this precise point.
>
> &nbsp;
>
> > ### Relation to "Penalizing Gradient Norm for Efficiently Improving Generalization in Deep Learning"
>
> The mentioned paper only uses gradient-norm penalty as a regularizer, and does not analyze it, let alone use the decomposition of the sharpness.
>
> &nbsp;
>
> > ### Complexity of combining Functional-SAM with preconditioning
>
> The implementation is a trivial and just a **one-liner change** in code (with negligible overhead). We don’t need to maintain any additional preconditioning statistics, but we can rely directly on that given by the base optimizer (Adam).
>
> &nbsp;
>
> > ### Code
>
> Yes, we aim to release code by the camera-ready stage.
>
> &nbsp;
>
>
> ---
>
> &nbsp;
>
> *Let us know if you have any further questions; if we have addressed your concerns we hope you will consider **revisiting your review score**.*
>
> &nbsp;
>
>
>
> &nbsp;
>
> [1] M. Andriushchenko and N. Flammarion. Towards understanding sharpness-aware minimization. ICML 2022.
>
> [2] P. D. Khanh et al. Fundamental Convergence Analysis of Sharpness-Aware Minimization. NeurIPS 2024
>
> [3] P. L. Bartlett et al. The dynamics of sharpness-aware minimization JMLR 2023.
>
> [4] Y. Dai et al. The crucial role of normalization in sharpness-aware minimization. NeurIPS 2023.
>
> [5] K. Ahn et al. How to escape sharp minima with random perturbations. ICML 2024.
>
> [6] https://proceedings.mlr.press/v202/agarwala23a
>
> [7] https://arxiv.org/abs/2203.02714
>
> [8] https://arxiv.org/abs/2503.02225

---

### Decision · Program_Chairs · 2025-05-01

**Decision:**

Accept (poster)

**Comment:**

Sharpness-aware minimization (SAM) is known to give great results when training vision models on small datasets but often performs poorly and even degrades performance when training LLMs. The work proposes a new method called Functional-SAM (along with preconditioned variants) which only perturbs the model and not the logits. Unlike regular SAM, the method shows consistent improvements when training language models at different scales.

Despite lower scores, reviewers appreciated several points in the paper. It is the first work to scale SAM to >1B models, considers the under explored SAM+LLM setting and even improves SAM to work for LLMs for the first time.

Several concerns were pointed out by reviewers, such as 1) lack of comparison to other SAM baselines, 2) additional overhead of SAM and 3) lack of theoretical advantages of functional SAM.  While reviewers did not change their score, I found most of these points to be addressed well in the rebuttal. There are ways to overcome the additional overheads, and even more computationally expensive methods could be interesting in view of scaling laws that also take the amount of data used into account.

Overall I believe the strong points outweigh the concerns raised by the reviewers. It is important that SAM research moves past convolutional networks on CIFAR and this work makes an important step in that direction and will be of great interest to the community. For the final version, I encourage the authors to take the reviewers suggestions into account, and a comparison of the Zoo of existing SAM variants on (perhaps a small) language model will be very valuable.